# PISCES: ANNOTATION-FREE TEXT-TO-VIDEO POST-TRAINING VIA BI-OBJECTIVE OT-ALIGNED REWARDS

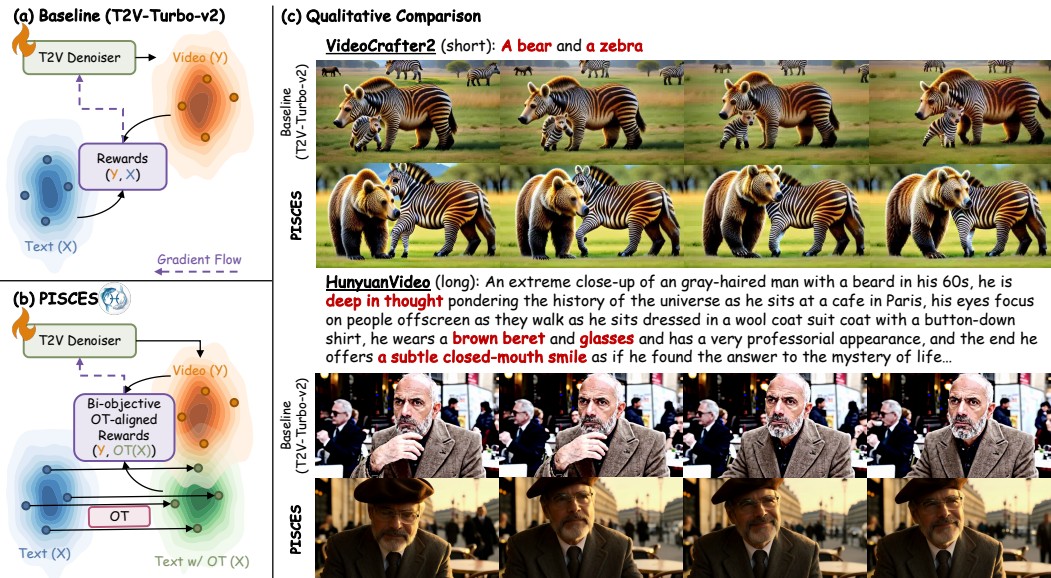

Figure 1: (a) Baseline (T2V-Turbo-v2) defines rewards over pre-trained VLM text-video embeddings, which suffer from distributional misalignment. (b) PISCES T2V post-training addresses this by formulating reward supervision over an OT-aligned embedding space. We propose a novel Bi-objective OT-aligned Rewards module that aligns text embeddings to the video space, enabling effective visual and semantic alignment. (c) Compared to the baseline, PISCES improves visual quality (temporal coherence, photorealism) and semantic fidelity (object count, attributes) on both short-video (VideoCrafter2) and long-video (HunyuanVideo) generation.

## ABSTRACT

Text-to-video (T2V) generation aims to synthesize videos with high visual quality and temporal consistency that are semantically aligned with input text. Reward-based post-training has emerged as a promising direction to improve the quality and semantic alignment of generated videos. However, recent methods either rely on large-scale human preference annotations or operate on misaligned embeddings from pre-trained vision-language models, both leading to limited scalability or suboptimal supervision. We present PISCES, an annotation-free post-training algorithm that addresses these limitations via a novel Bi-objective Optimal Transport (OT)-aligned Rewards module. To align reward signals with human judgment, PISCES uses OT to bridge text and video embeddings at both the distribution and discrete token levels, enabling reward supervision to fulfill two objectives: (i) a Distributional OT-aligned Quality Reward that captures overall visual quality and temporal coherence; and (ii) a Discrete Token-level OT-aligned Semantic Reward that enforces semantic, spatio-temporal correspondence between text and video tokens. To our knowledge, PISCES is the first to improve annotation-free reward supervision in generative post-training through the lens of OT. Experiments on both short-video and long-video generation show that PISCES outperforms both annotation-free and annotation-based methods on VBench across Quality and Semantic scores, with human preference studies further validating its effectiveness. We also show that the Bi-objective OT-aligned Rewards module is compatible with multiple optimization paradigms, including direct backpropagation and reinforcement learning fine-tuning.

## 1 INTRODUCTION

Text-to-video (T2V) generation (Kong et al., 2025; Marwah et al., 2017) aims to synthesize videos from textual descriptions such that the videos appear realistic, temporally consistent, and accurately reflect the prompt. T2V has broad applications in multimedia content creation, robotics, and accessibility. While the performance of T2V models is inherently subjective and judged by human preferences, recent benchmarks (Huang et al., 2024) formalize evaluation along two main dimensions – *Quality score*, accounting for the visual quality and temporal consistency; and *Semantic score*, factoring the correspondence of generated videos to text prompts.

Rapid advances in diffusion and flow matching models (Podell et al., 2024; Esser et al., 2024) and Vision-Language Models (VLMs) (Chung et al., 2023; Chen et al., 2023; Kelly et al., 2024) have enabled the development of recent T2V models (Pika Labs, 2023; RunwayML, 2024; He et al., 2023; Kong et al., 2025). To further improve existing T2V models (He et al., 2023; Kong et al., 2025), particularly in terms of video-text misalignment in the denoisers, reward-based post-training (Li et al., 2025; Liu et al., 2025a) has been introduced that provides additional supervision via specially designed rewards to the denoiser.

Reward-based T2V post-training methods can be either Annotation-based or Annotation-free. Annotation-based approaches (Liu et al., 2025a; Yang et al., 2025; Wang et al., 2025) collect large-scale human preference datasets, where annotators judge generated video pairs on quality and semantics, which are used to train a reward model or via Direct Preference Optimization (DPO) (Rafailov et al., 2023; Wallace et al., 2024) for post-training. Although effective and serving as existing SoTA, these annotation-based methods cannot easily scale because they need high-quality preference-based annotations. Another line of work explores Annotation-free rewards, where supervision is derived from pre-trained VLMs rather than human labels (Li et al., 2024a; 2025). While these approaches do not need large-scale human annotations, their performance is not on par with the Annotation-based techniques. We aim to achieve the best of both worlds by asking: *Can an annotation-free T2V post-training method match, or even outperform, annotation-based approaches*?

From a review of annotation-free approaches, we identify reliance on pre-trained vision–language models (VLMs) for reward supervision as a key limitation. VLMs are trained with non-distributional objectives, such as pointwise matching (Chen et al., 2020) and contrastive learning (Radford et al., 2021), that may not adequately align text with the real-video distribution, consistent with the patterns in Table 5 and Figure 6. This results in both quality and semantic issues, as shown in Figure 1c, such as failure to ensure the correct number and attributes of objects (*e.g.*, "a zebra and a bear", "wearing a brown beret and glasses") or failing to capture motion descriptors (*e.g.*, "closed-mouth smile").

We posit that, for annotation-free reward supervision to mimic human preferences, the real-video distribution must be better aligned with the text distribution, which represents the space of human instructions/preferences, without altering the video distribution's semantic structure, and the derived rewards should reflect human judgments of T2V outputs on the bi-objective of quality and semantics. We introduce PISCES[1], an annotation-free T2V post-training algorithm that includes a novel **Bi-objective Optimal Transport-aligned Rewards** module (Figure 1b). Leveraging Optimal Transport (OT) (Villani, 2009; Cuturi, 2013), we tailor PISCES specifically for T2V post-training by enhancing text-video alignment at both the distribution and the token level to simultaneously improve both the visual quality and semantic consistency. For this, the Rewards module comprises: (i) a **Distributional OT-aligned Quality Reward**, which learns a distributional-OT map to transform text embeddings into the real-video embedding space while preserving their internal structure and enforcing temporal consistency and visual quality; and (ii) a **Discrete Token-level OT-aligned Semantic Reward**, which constructs a spatio-temporal cost matrix over text and video tokens and solves a partial OT problem with an entropic Sinkhorn solver (Cuturi, 2013), to supervise correspondence by aligning text tokens with the most semantically, spatially, and temporally consistent video regions.

We validate PISCES on both short-video (VideoCrafter2 (He et al., 2023)) and long-video (Hunyuan-Video (Kong et al., 2025)) generator (Figure 1c) via VBench (Huang et al., 2024) as well as human evaluation. We show that our Bi-objective OT-aligned Rewards module is applicable across different optimization paradigms, including direct backpropagation (gradient backpropagation through reward

---

[1]In astrology, Pisces is symbolized by two fish, signifying balance and integration of multiple realms. PISCES echoes this by aligning text and video through complementary rewards for quality and semantics.

models) and reinforcement learning (RL) fine-tuning (GRPO (Guo et al., 2025; Xue et al., 2025)). In doing so, we find that PISCES can significantly outperform all existing reward-based post-training approaches (both Annotation-free and Annotation-based). Through careful realignment of the text-video space, PISCES demonstrates the ability to outperform Annotation-based approaches while remaining Annotation-free, making it a much stronger alternative at scale. Our key contributions are:

- We introduce PISCES, a novel annotation-free post-training framework for T2V generation. For the first time, we identify a core bottleneck in existing VLM-based rewards–operating on misaligned text-video embeddings–and address this by leveraging OT to align embeddings, enabling reward supervision in a semantically meaningful, structure-preserving space.

- PISCES defines a novel Bi-objective OT-aligned Rewards module comprising: (1) a *Distributional OT-aligned Quality Reward*, capturing overall visual quality and temporal consistency; and (2) a *Discrete Token-level OT-aligned Semantic Reward*, targeting localized text-video alignment for semantic consistency.

- PISCES outperforms both annotation-based and annotation-free T2V post-training methods on both Semantic and Quality Scores for short and long video generation, as validated by automatic metrics and human evaluations. We show the OT-aligned Rewards module is applicable to multiple optimization strategies.

## 2 RELATED WORK

**Reward-based Post-Training for T2V.** In the image domain, reward models such as HPSv2 (Wu et al., 2023), ImageReward (Xu et al., 2023), and PickScore (Kirstain et al., 2023) have proven effective for aligning generations with text prompts. Extending to video, annotation-based approaches train on large-scale human preference datasets, including VideoReward (Liu et al., 2025a), IPO (Yang et al., 2025), and UnifiedReward (Wang et al., 2025). While effective, these methods incur high annotation costs and suffer from limited scalability. Orthogonally, annotation-free methods leverage pre-trained VLMs such as ViCLIP (Wang et al., 2023b) and InternVideo2 (Wang et al., 2024), with cosine-similarity rewards adopted in T2V-Turbo (Li et al., 2024a), T2V-Turbo-v2 (Li et al., 2025), while InstructVideo (Yuan et al., 2024) still relies on image-text rewards. However, these rewards operate on misaligned text–video embedding spaces, which reduces their effectiveness to perform on par with annotation-based methods. We identify this reward misalignment as the core bottleneck in T2V post-training and propose PISCES, the first to explore aligning embeddings via OT in generative post-training. Our framework introduces Bi-objective distributional and token-level OT-aligned rewards, enabling scalable and effective annotation-free supervision.

**Optimal Transport.** Optimal Transport (OT) provides a principled framework for aligning probability distributions and has been widely applied in machine learning tasks such as domain adaptation (Katageri et al., 2024; Le Duy et al., 2024), generative modeling (Tong et al., 2024; Li et al., 2023), and cross-modal retrieval (Han et al., 2024; Izquierdo & Civera, 2024). Neural Optimal Transport (NOT)(Korotin et al., 2023) further offers a scalable alternative by learning explicit transport maps via neural networks. Recent works have also leveraged discrete OT for alignment in vision tasks: Xie et al. (2025) formulate disentangled OT for visual–concept relations, while Liu et al. (2025b) integrates OT into query reformation for temporal action localization. Other efforts include HOTS3D (Li et al., 2024b), which aligns text and image features using spherical OT, and OT-CLIP (Shi et al., 2024), which reframes CLIP training and inference as OT problems. Despite these advances, prior work has not explored OT in the context of reward modeling for generative T2V post-training. For the first time, PISCES identifies misalignment text-video embeddings in pre-trained VLMs for annotation-free T2V reward formulation, and introduces a Bi-objective OT-aligned Rewards module jointly capturing distributional alignment and discrete token-level text-video correspondence.

## 3 METHOD

Figure 2 provides an overview of PISCES's novel Bi-objective OT-aligned Rewards module and T2V post-training algorithm.

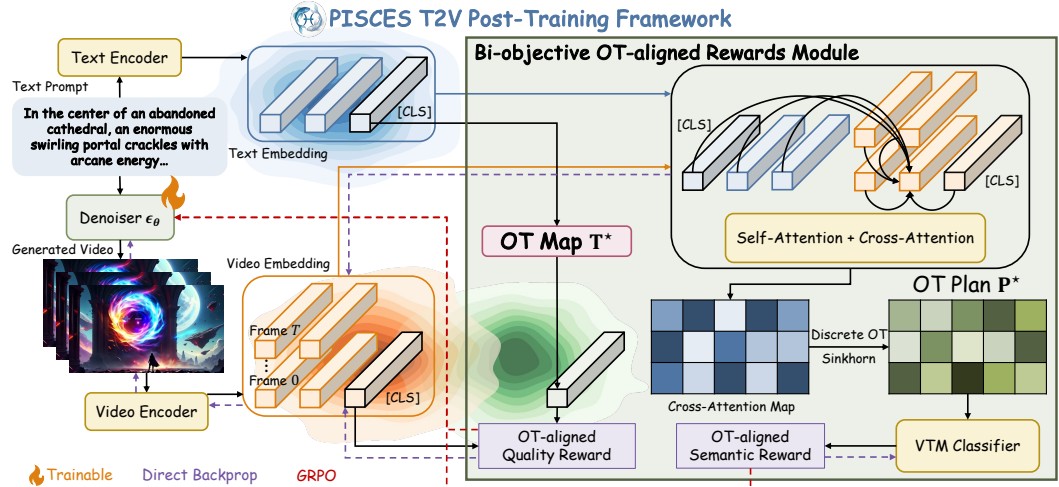

Figure 2: PISCES **T2V Post-Training.** We introduce a Bi-objective OT-aligned Rewards module: (i) a distributional OT map $\mathbf{T}^\star$ for Quality Reward via [CLS] representation similarity, and (ii) a discrete OT plan $\mathbf{P}^\star$ with spatio-temporal constraints for Semantic Reward via a Video-Text Matching (VTM) classifier. The rewards module provides supervision for fine-tuning the T2V denoiser and is applicable with direct backpropagation and RL fine-tuning (GRPO).

## 3.1 DISTRIBUTIONAL OT-ALIGNED QUALITY REWARD

When humans evaluate the *quality* of a generated video, they attend to global properties such as realism, motion coherence, and overall visual consistency – essentially asking whether the video could plausibly belong to the distribution of real-videos. To mimic this process in an annotation-free manner, we align text embeddings onto the manifold of real-video embeddings before defining the reward.

For this, we formulate the distributional alignment as a Monge–Kantorovich Optimal Transport (OT) problem. OT (Villani, 2009) provides a principled framework for aligning two probability distributions $\mu \in \mathcal{P}(\mathcal{Y})$ and $\nu \in \mathcal{P}(\mathcal{X})$ by finding a transport map $\mathbf{T} : \mathcal{Y} \to \mathcal{X}$ that pushes $\mu$ to $\nu$ (i.e., $\mathbf{T}_\sharp \mu = \nu$) while minimizing a transport cost $\mathbf{c}(\mathbf{y}, \mathbf{x})$. Given text embeddings $\mathcal{Y}$ and real-video embeddings $\mathcal{X}$ extracted from a pre-trained VLM, we train an OT map $\mathbf{T} : \mathcal{Y} \to \mathcal{X}$ using NOT (Korotin et al., 2023) with the objective:

$$\sup_f \inf_{\mathbf{T}} \int_{\mathcal{X}} f(\mathbf{x}) d\nu(\mathbf{x}) + \int_{\mathcal{Y}} \left( \mathbf{c}(\mathbf{y}, \mathbf{T}(\mathbf{y})) - f(\mathbf{T}(\mathbf{y})) \right) d\mu(\mathbf{y}), \tag{1}$$

where $\mathbf{c}(\mathbf{y}, \mathbf{x}) = \|\mathbf{y} - \mathbf{x}\|^2$ is the transport cost. We implement this via iterative optimization of the transport map $\mathbf{T}_\psi$ and potential function $f_\omega$ parameterized by neural networks, as shown in Algorithm 2 in Appendix B. The resulting OT-aligned embeddings $\mathbf{T}^\star(\mathbf{y})$ reduce distributional mismatch while preserving the semantic structure of the embedding space (see Table 5 and Figure 6).

Once the text distribution is aligned with the real-video distribution, comparing an OT-aligned text embedding $\mathbf{T}^\star(\mathbf{y})$ with a generated video embedding $\hat{\mathbf{x}}$ becomes equivalent to comparing a real-video embedding $\mathbf{x}^{\text{real}}$ with $\hat{\mathbf{x}}$. The OT map thus projects text embeddings into the real-video manifold, making $\mathbf{T}^\star(\mathbf{y})$ a proxy for $\mathbf{x}^{\text{real}}$. With this intuition, we define the Quality Reward (Figure 2) as the cosine similarity between global representation [CLS] tokens:

$$\mathcal{R}_{\text{OT-quality}} = \frac{\mathbf{T}^\star(\mathbf{y}_{[\text{CLS}]})^T \cdot \hat{\mathbf{x}}_{[\text{CLS}]}}{\|\mathbf{T}^\star(\mathbf{y}_{[\text{CLS}]})\| \|\hat{\mathbf{x}}_{[\text{CLS}]}\|} \approx \frac{(\mathbf{x}_{[\text{CLS}]}^{\text{real}})^T \cdot \hat{\mathbf{x}}_{[\text{CLS}]}}{\|\mathbf{x}_{[\text{CLS}]}^{\text{real}}\| \|\hat{\mathbf{x}}_{[\text{CLS}]}\|}. \tag{2}$$

Cosine similarity provides a natural coherence signal by comparing the *direction* of embeddings, making it robust to scale or style differences while capturing structural consistency. After OT projects text embeddings into the real-video manifold, cosine becomes a geometry-respecting measure of quality, evaluating whether generated videos point in the same "quality direction" as real ones.

## 3.2 DISCRETE TOKEN-LEVEL OT-ALIGNED SEMANTIC REWARD

When judging semantic fidelity in T2V, humans implicitly ask whether the prompt's key words are actually reflected in the generated video. To mimic this in T2V post-training, we introduce a token-level reward based on Partial Optimal Transport (POT).

To facilitate a strong semantic alignment, we integrate discrete POT into the cross-attention layers of InternVideo2. Vanilla cross-attention, however, often fails to capture precise multimodal correspondences: it operates directly on misaligned embeddings and distributes attention diffusely across irrelevant patches, as seen in Figure 5. Without a mechanism to enforce selective, structured grounding, important tokens may fail to connect to the right visual regions. To address this problem, we design a novel mechanism which augments attention with a POT-guided transport plan that enforces semantic, temporal, and spatial consistency between text and video tokens. For each cross-attention head, we construct a cost matrix between text tokens $\mathbf{y}$ and video patch tokens $\hat{\mathbf{x}}$ comprising three components specifically designed for T2V rewards:

**Semantic similarity:** $1 - \cos(\mathbf{y}_i, \hat{\mathbf{x}}_j)$, encouraging tokens with similar meaning to align.

**Temporal constraint:** $|\tau(\mathbf{y}_i) - t_j|$, where $\tau(\mathbf{y}_i) = \sum_k \mathbf{A}_{ik} * t_k$ is the expected frame index of text token $i$ under the attention distribution $\mathbf{A}$, and $t_j$ is the frame index of video patch $j$.

**Spatial constraint:** $|\pi(\mathbf{y}_i) - s_j|_2$, where $\pi(\mathbf{y}_i) = \sum_k \mathbf{A}_{ik} * s_k$ is the expected 2D position of text token $i$ (under attention $\mathbf{A}$) and $s_j$ is the spatial coordinate of video patch $j$ on the frame grid.

The final cost matrix is: $\mathbf{C}_{ij} = \text{semantic}(i,j) + \gamma \cdot \text{temporal}(i,j) + \eta \cdot \text{spatial}(i,j)$, with $\gamma, \eta$ balancing temporal and spatial penalties. We then solve a partial OT problem on this cost matrix $\mathbf{C}$ via an entropic Sinkhorn solver (Cuturi, 2013) with fraction-of-mass $m = 0.9$, as shown in Algorithm 3. This produces a transport plan $\mathbf{P}^\star$ that softly matches each text token to a subset of video tokens, rather than forcing full mass transport. To integrate this into InternVideo2, we propose to inject $\mathbf{P}^\star$ into the vanilla attention $\mathbf{A}$ via log-space fusion, a lightweight, differentiable mechanism. This yields updated cross-attention probabilities $\tilde{\mathbf{A}}$ that combines standard attention with POT-guided structure. Formally, the updated cross-attention map is:

$$\tilde{\mathbf{A}} \propto \exp\big(\log(\mathbf{A} + \varepsilon) + \log(\mathbf{P}^\star + \varepsilon)\big). \tag{3}$$

This fusion preserves differentiability through $\mathbf{A}$ while treating $\mathbf{P}^\star$ as a structural prior. Finally, the POT-refined features are passed into the pre-trained Video-Text Matching (VTM) classifier of InternVideo2, which outputs two logits for positive and negative matches. The positive logit after softmax ($\text{idx} = 1$) provides the Semantic Reward:

$$\mathcal{R}_{\text{OT-semantic}} = \texttt{softmax}\left(\texttt{VTM}\left[\tilde{\mathbf{A}} \cdot \hat{\mathbf{x}}\right]\right)_{\text{idx}=1}. \tag{4}$$

Our discrete POT-based Semantic Reward captures human selectivity: not every word needs to be grounded, and important tokens are matched to relevant patches. The spatio-temporal cost further constrains *where* and *when* content should appear. Together, this provides a reward signal that evaluates text–video correspondence in a human-aligned way.

## 3.3 POST-TRAINING

**Direct Backpropagation.** We integrate the Bi-objective OT-aligned Rewards module into consistency distillation (Song et al., 2023; Wang et al., 2023c) for efficient refinement, optimizing the denoiser:

$$\mathcal{L}_{\text{direct}} = \mathcal{L}_{\text{CD}}(\boldsymbol{\theta}, \boldsymbol{\theta}^-; \phi) - \mathcal{R}_{\text{OT-quality}} - \mathcal{R}_{\text{OT-semantic}}. \tag{5}$$

**GRPO.** At each step, given a prompt $\mathbf{y}$, we sample a group of videos using SDEs $\{\mathbf{x}_0^1, \mathbf{x}_0^2, \ldots, \mathbf{x}_0^G\}$ from the video denoiser $\pi_{\theta_{\text{old}}}$, and optimize the policy model $\pi_\theta$ by minimizing the objective function (Guo et al., 2025; Xue et al., 2025):

$$\mathcal{L}_{\text{GRPO}} = \mathbb{E}_{\substack{\{\mathbf{x}_0^i\}_{i=1}^G \sim \pi_{\theta_{\text{old}}}(\cdot|\mathbf{y}) \\ \mathbf{a}_{t,i} \sim \pi_{\theta_{\text{old}}}(\cdot|\mathbf{s}_{t,i})}} \left[ \frac{1}{G} \sum_{i=1}^G \frac{1}{T} \sum_{t=1}^T \max\left(-\rho_{t,i} A_i, -\text{clip}\left(\rho_{t,i}, 1-\epsilon, 1+\epsilon\right) A_i\right) \right], \tag{6}$$

where $\rho_{t,i} = \frac{\pi_\theta(\mathbf{a}_{t,i}|\mathbf{s}_{t,i})}{\pi_{\theta_{old}}(\mathbf{a}_{t,i}|\mathbf{s}_{t,i})}$ and $A_i = \frac{r_i - \text{mean}(\{r_1, r_2, \ldots, r_G\})}{\text{std}(\{r_1, r_2, \ldots, r_G\})}$ is the advantage function computed using a group of rewards $\{r_1, r_2, \ldots, r_G\}$ with our Bi-objective OT-aligned Rewards module.

---

**Algorithm 1** Post-Training with Bi-objective OT-aligned Rewards module

---

**Require:** Pre-trained denoiser $\epsilon_\theta$; data $\mathbf{p}_{\text{data}}$; ODE solver $\Phi$; skipping interval $k$; distance $d(\cdot,\cdot)$; $\theta^- \leftarrow \theta$
**Ensure:** $\epsilon_\theta$ converges and minimizes $\mathcal{L}_{\text{total}}$
   **while** not converged **do**
        Sample video-text $(\mathbf{x}_{\text{video}}, \mathbf{y}_{\text{text}}) \sim \mathbf{p}_{\text{data}}$, $n \sim \mathcal{U}[1, N-k]$
        $\mathbf{z}_0 \leftarrow \mathcal{E}(\mathbf{x}_{\text{video}})$                 $\triangleright$ Encode $\mathbf{x}_{\text{video}}$ to latent space $\mathbf{z}_0$; Extract text embedding $\mathbf{y}$ from $\mathbf{y}_{\text{text}}$
        Add noise to latent $\mathbf{z}_{t_{n+k}} \sim \mathcal{N}\left(\alpha(t_{n+k})\mathbf{z}_0; \beta^2(t_{n+k})\mathbf{I}\right)$
        $\hat{\mathbf{z}}_{t_n}^\phi \leftarrow \mathbf{z}_{t_{n+k}} + (t_n - t_{n+k})\Phi(\mathbf{z}_{t_{n+k}}, t_{n+k}; \phi)$        $\triangleright$ Perform ODE solver from $t_{n+k} \rightarrow t_n$
        $\mathcal{L}_{\text{CD}}(\theta, \theta^-; \phi) \leftarrow d\left(g_\theta(\mathbf{z}_{t_{n+k}}, t_{n+k}), g_{\theta^-}(\hat{\mathbf{z}}_{t_n}^\phi, t_n)\right)$    $\triangleright$ Compute Consistency Distillation loss
        $\hat{\mathbf{z}}_0^\phi \leftarrow \mathbf{z}_{t_{n+k}} - \int_0^{t_{n+k}}\left(\gamma(t)\mathbf{z}_t + \frac{1}{2}\sigma^2(t)\epsilon_\theta(\mathbf{z}_t, \mathbf{y}, t)\right)\mathrm{d}t$   $\triangleright$ Single-step ODE solver from $t_{n+k} \rightarrow 0$
        $\hat{\mathbf{x}}_0 \leftarrow \mathcal{D}(\hat{\mathbf{z}}_0^\phi)$                          $\triangleright$ Decode $\hat{\mathbf{z}}_0^\phi$ to pixel space $\hat{\mathbf{x}}_0$
        Extract video embedding $\hat{\mathbf{x}}$ from video $\hat{\mathbf{x}}_0$ and compute rewards using $\hat{\mathbf{x}}$ and $\mathbf{y}$ with OT
        $\mathcal{L}_{\text{direct}} = \mathcal{L}_{\text{CD}}(\theta, \theta^-; \phi) - \mathcal{R}_{\text{OT-quality}} - \mathcal{R}_{\text{OT-semantic}}$ or $\mathcal{L}_{\text{totalGRPO}} = \mathcal{L}_{\text{CD}}(\theta, \theta^-; \phi) + \mathcal{L}_{\text{GRPO}}$
        Backward $\mathcal{L}_{\text{direct}}$ or $\mathcal{L}_{\text{totalGRPO}}$ to update $\theta$ and $\theta^- \leftarrow \texttt{stop\_grad}(\lambda\theta + (1-\lambda)\theta^-)$
   **end while**

---

During post-training, we use LoRA (Hu et al., 2022), freezing all parameters except the denoiser. Algorithm 1 summarizes this procedure: we generate videos using Euler ODE for direct backprop and SDE for GRPO, compute the Bi-objective OT-aligned Rewards, and optimize the denoiser.

## 4 EXPERIMENTS

### 4.1 EXPERIMENTAL SETTING

**Implementation Details.** We validate `PISCES` by post-training VideoCrafter2 (He et al., 2023) (2s @ 8FPS) and HunyuanVideo (Kong et al., 2025) (5s @ 25FPS), representing short- and long-video settings. We use base text-video embeddings from InternVideo2 (Wang et al., 2024) for OT-based reward alignment. Post-training is performed on $8\times$ A100 80GB GPUs for 2 days with a learning rate of $1e-6$, batch size 1, and accumulation 32 (direct backprop) or 4 days for GRPO (includes time for intermediate inference and visualization, actual training time is $\approx 30$ hours for direct backpropagation and $\approx 78$ hours for GRPO). Both the OT map $\mathbf{T}_\psi$ and critic $f_\omega$ are 3-layer MLPs with ReLU activations and LayerNorm. To train the Neural OT (NOT) map, we use video-text pairs with 8-frame clips, extracted using frozen InternVideo2 (Wang et al., 2024), and train on a single A100 GPU for one day, equivalent to 24 A100 GPU-hours.

**Datasets.** Following T2V-Turbo-v2 (Li et al., 2025), we construct a balanced dataset mixing WebVid10M (Bain et al., 2021) and VidGen-1M (Tan et al., 2024). To ensure consistency across models, we sample 2s clips at 8FPS for VideoCrafter2 (short-video) and 5s clips at 25FPS for HunyuanVideo (long-video). We resize frames to $512 \times 320$ for VideoCrafter2 and $848 \times 480$ for HunyuanVideo before training.

**Evaluation Metrics.** We evaluate `PISCES` with VBench (Huang et al., 2024), benchmarking T2V generation across 16 dimensions summarized into a *Quality Score* (visual fidelity and temporal coherence, *e.g.*, subject/background consistency, motion smoothness) and a *Semantic Score* (fine-grained alignment with prompts, *e.g.*, object presence, spatial relations, action correctness). The *Total Score*, a weighted sum of the two, offers a holistic measure of video fidelity and semantic alignment. We also conduct human evaluations on 400 prompts following VideoReward (Liu et al., 2025a).

### 4.2 COMPARISON WITH EXISTING METHODS

**Automatic Evaluation on VBench.** Table 1 compares `PISCES` with existing T2V post-training methods (Li et al., 2024a; 2025; Liu et al., 2025a;c; Wang et al., 2025) on both short-video (VideoCrafter2) and long-video (HunyuanVideo) generation. We observe that `PISCES` significantly outperforms both annotation-based and annotation-free approaches, achieving the highest scores across all metrics – Total, Quality, and Semantic. Table 2 provides a broader comparison with recent T2V models, both open-source (Wang et al., 2023a; Kong et al., 2025; He et al., 2023; Zhang et al., 2024) and closed-source (Pika Labs, 2023; KlingAI, 2025; RunwayML, 2024). We can again observe that, when post-trained with `PISCES`, HunyuanVideo outperforms existing T2V models on automatic VBench

Table 1: **Automatic VBench comparison on VideoCrafter2 and HunyuanVideo**. PISCES significantly outperforms existing reward-based T2V post-training methods across all scores. Reproduced without motion guidance for fair comparison. Additional analysis in Appendix G

| Models | VideoCrafter2 (He et al., 2023) | | | HunyuanVideo (Kong et al., 2025) | | |
|---|---|---|---|---|---|---|
| | Total | Quality | Semantic | Total | Quality | Semantic |
| Vanilla | 80.44 | 82.20 | 73.42 | 83.24 | 85.09 | 75.82 |
| VCM (Wang et al., 2023c) | 73.97 ↓6.47 | 78.54 ↓3.66 | 55.66 ↓17.8 | 81.77 ↓1.47 | 84.60 ↓0.49 | 70.49 ↓5.33 |
| T2V-Turbo (Li et al., 2024a) | 81.01 ↑0.57 | 82.57 ↑0.37 | 74.76 ↑1.34 | 83.86 ↑0.62 | 85.57 ↑0.48 | 77.00 ↑1.18 |
| T2V-Turbo-v2* (Li et al., 2025) | 81.87 ↑1.43 | 83.26 ↑1.06 | 76.30 ↑2.88 | 84.25 ↑1.01 | 85.93 ↑0.84 | 77.52 ↑1.70 |
| VideoReward-DPO (Liu et al., 2025a) | 80.75 ↑0.31 | 82.11 ↓0.09 | 75.29 ↑1.87 | 83.54 ↑0.30 | 85.02 ↓0.07 | 77.63 ↑1.81 |
| VideoDPO (Liu et al., 2025c) | 81.93 ↑1.49 | 83.07 ↑0.87 | 77.38 ↑3.96 | 84.13 ↑0.89 | 85.71 ↑0.62 | 77.83 ↑2.01 |
| UnifiedReward (Wang et al., 2025) | 81.43 ↑0.99 | 83.26 ↑1.06 | 74.12 ↑0.70 | 83.80 ↑0.56 | 85.46 ↑0.37 | 77.15 ↑1.33 |
| PISCES | **82.75 ↑2.31** | **84.05 ↑1.85** | **77.54 ↑4.12** | **85.45 ↑2.21** | **86.73 ↑1.64** | **80.33 ↑4.51** |

Table 2: **Automatic Evaluation on VBench**. We compare different T2V models across Quality, Semantic, and Total Scores. HunyuanVideo, post-trained with PISCES, performs the best on all scores across all models.

| Metric | ModelScope (Wang et al., 2023a) | Show-1 (Zhang et al., 2024) | Pika-1.0 (Pika Labs, 2023) | Gen-3 (RunwayML, 2024) | Kling (KlingAI, 2025) | VideoCrafter2 (He et al., 2023) | HunyuanVideo (Kong et al., 2025) | PISCES | |
|---|---|---|---|---|---|---|---|---|---|
| | | | | | | | | VideoCrafter2 | HunyuanVideo |
| Quality Score | 78.05 | 80.42 | 82.92 | 84.11 | 83.39 | 82.20 | 85.09 | 83.73 | **86.73** |
| Semantic Score | 66.54 | 72.98 | 71.77 | 75.17 | 75.68 | 73.42 | 75.82 | 77.63 | **80.33** |
| Total Score | 75.75 | 78.93 | 80.69 | 82.32 | 81.85 | 80.44 | 83.24 | 82.51 | **85.45** |

evaluation. This highlights the benefit of our OT formulation to align the text-video embeddings in an off-the-shelf pre-trained VLM, which strengthens supervision for the T2V post-training Rewards module. This also highlights the effectiveness of our proposed OT-aligned Quality and Semantic Rewards in performing optimal T2V post-training. We also show PISCES's OT-aligned Rewards module is applicable to different optimizations in Appendix D.

**Human Evaluation.** Following VideoReward (Liu et al., 2025a), we conduct a human preference study with 400 prompts evaluating along three dimensions: visual quality, motion quality, and text alignment. For each prompt, we generate videos using pre-trained HunyuanVideo, post-trained Hunyuan-Video with T2V-Turbo-v2 (Li et al., 2025), VideoReward-DPO Liu et al. (2025a), and PISCES. We ask the participants three questions: (1) Which video is better aligned with the text prompt? (2) Which video has better visual quality? and (3) Which video has better motion quality? We collect responses from 85 participants, comparing PISCES against Hunyuan-Video, T2V-Turbo-v2, and VideoReward-DPO

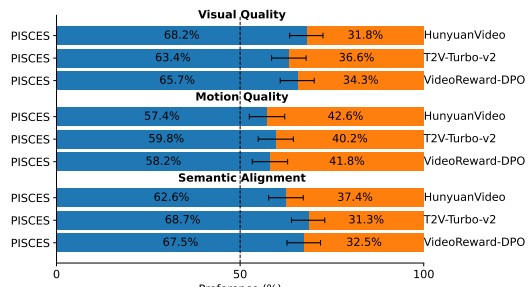

Figure 3: **Human preference study.** PISCES outperforms HunyuanVideo, T2V-Turbo-v2, VideoReward-DPO in visual quality, motion quality and semantic alignment, validating its effectiveness in improving T2V.

on Quality and Semantic dimensions. As shown in Figure 3, PISCES is preferred over pre-trained HunyuanVideo, T2V-Turbo-v2 and VideoReward-DPO in terms of visual quality, motion quality, and semantic alignment. These results indicate PISCES's Bi-objective OT-aligned Rewards effectively enhance both quality and text-video semantic consistency, leading to more visually appealing and semantically coherent generations. Further, experiments on evaluation prompts from VBench (Huang et al., 2024) and VideoReward (Liu et al., 2025a) imply OT-aligned Rewards in PISCES's T2V post-training can generalize the performance of T2V generation on out-of-domain prompts because we train using WebVid10M (Bain et al., 2021) and VidGen-1M (Tan et al., 2024).

**Qualitative Comparison.** To further assess the effectiveness of PISCES, we visually compare videos generated by PISCES against other methods in Figure 4. Given the text prompt, we observe PISCES generates videos with improved semantic fidelity and visual coherence. Compared to baselines, PISCES is better at preserving fine-grained semantic details, such as the reflection effects on the wet pavement and the vibrant color contrast in the scene (w.r.t. Semantic Score in VBench). The generated subject is also more globally consistent across frames, reducing temporal flicker and maintaining a stable appearance (w.r.t. to Quality Score in VBench). These results align with our quantitative findings from the previous section, show the effectiveness of PISCES's Bi-objective OT-aligned Rewards to enhance both visual quality and text-video alignment in T2V models.

**Prompt**: A stylish woman walks down a Tokyo street filled with warm glowing neon and animated city signage. She wears a black leather jacket, a long red dress, and black boots, and carries a black purse. She wears sunglasses and red lipstick. She walks confidently and casually. The street is damp and reflective, creating a mirror effect of the colorful lights. Many pedestrians walk about.

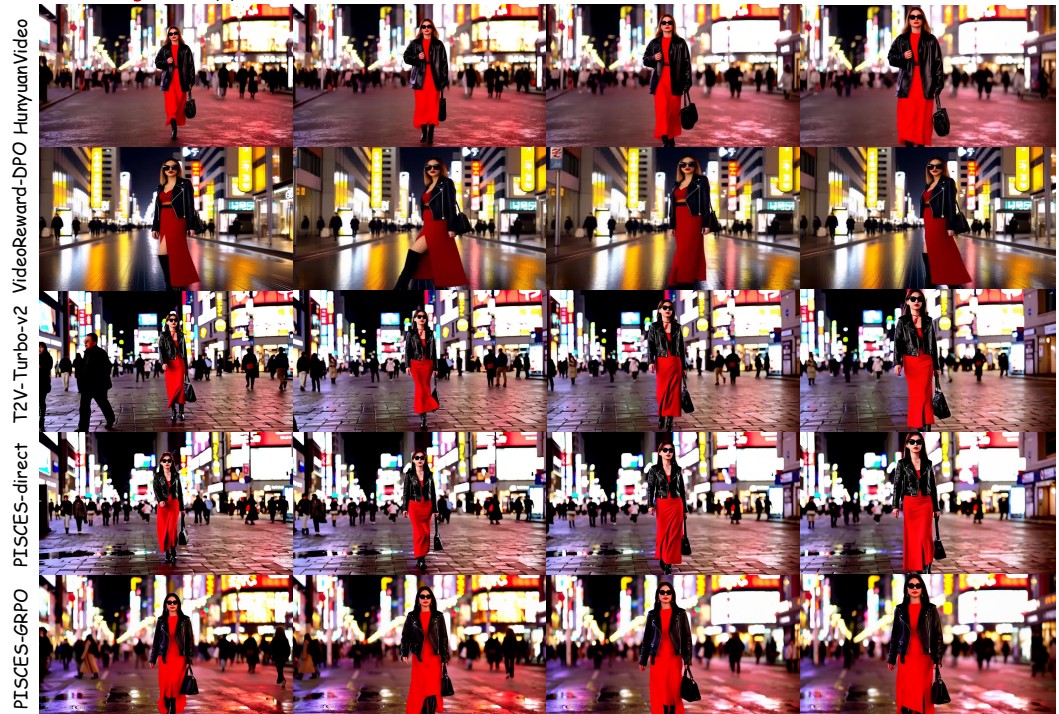

Figure 4: Qualitative comparison of T2V models. PISCES produces videos with better semantic fidelity and visual quality, accurately capturing key details such as the reflective wet pavement and vibrant neon lighting.

## 4.3 ABLATION STUDY

**Effectiveness of OT in Text-Video Alignment.** Table 3 (Rows 2 and 5) compares PISCES with and without OT on VideoCrafter2 to evaluate the impact of aligning the text and video distributions of the VLM via OT on T2V performance. Specifically, for Quality Reward, we replace the OT-aligned $\mathbf{T}^{\star}(\mathbf{y}_{[CLS]})$ term in Equation (2) with the raw $\mathbf{y}_{[CLS]}$. For Semantic Reward, we remove the discrete Partial OT attention map $\tilde{\mathbf{A}}$ in Equation (4) and uses regular cross-attention weights $\mathbf{A}$ between text and video tokens. We

Table 3: **Ablation Study.** OT alignment improves both Quality and Semantic Scores, while Quality Reward enhances visual quality and Semantic Reward improves text-video correspondence. Full PISCES achieves the **best** performance.

| Method | OT | $\mathcal{R}_{semantic}$ | $\mathcal{R}_{quality}$ | Total Score | Quality Score | Semantic Score |
|---|---|---|---|---|---|---|
| Vanilla (VideoCrafter2) | ✗ | ✗ | ✗ | 80.44 | 82.20 | 73.42 |
| PISCES w/o OT | ✗ | ✓ | ✓ | 81.92 | 83.44 | 75.82 |
| PISCES + $\mathcal{R}_{OT\text{-quality}}$ | ✓ | ✗ | ✓ | 82.21 | **83.77** | 75.97 |
| PISCES + $\mathcal{R}_{OT\text{-semantic}}$ | ✓ | ✓ | ✗ | 81.70 | 82.87 | 76.99 |
| PISCES Full | ✓ | ✓ | ✓ | **82.51** | 83.73 | **77.63** |

observe OT plays a crucial role in improving both the Quality and Semantic Scores. PISCES achieves a Semantic Score of 77.63, outperforming PISCES without OT (75.82) and demonstrating that aligning text and video distributions before leveraging them for T2V post-training reward function significantly enhances semantic correspondence. OT alignment also improves Quality Score (83.73 vs. 83.44), further validating the importance of OT in structuring the feature space for improved video generation. These results confirm that aligning text and video embeddings distribution with OT provides stronger supervision compared to existing post-training methods, which rely on off-the-shelf VLM embeddings that might be suboptimal for the task.

**Impact of OT-aligned Quality and Semantic Rewards.** Table 3 analyzes the contribution of each OT-aligned Reward. Using the Quality Reward alone (Row 3) improves the VBench Quality Score from 82.20 to 83.77 (Row 1 baseline), demonstrating its effectiveness in enhancing global coherence, visual quality, and subject consistency. In contrast, training with only the Semantic Reward (Row 4) substantially boosts the Semantic Score from 73.42 to 76.99, highlighting its role in capturing fine-

grained text–video alignment such as object presence, actions, and temporal style. When we combine both rewards (Row 5), `PISCES` achieves the best overall performance, yielding improvements in both Quality and Semantic Scores compared to single-reward variants. This confirms the complementary nature of the two Rewards and the benefit of operating jointly in the `PISCES`'s OT-aligned space. For full comparisons across all 16 VBench dimensions, please refer to Appendix E.

**Group Relative Reward Fusion**. In our default direct backpropagation setup, we equally weight the consistency loss and reward signals. Moreover, we explore an adaptive weighting strategy using the Group-Relative Reward formulation from GRPO (Guo et al., 2025; Xue et al., 2025)–normalizing reward values across generations for the same prompt (subtracting the mean and dividing by standard deviation). We apply this normalization in the direct backpropagation setting without using RL. Results on HunyuanVideo (Kong et al., 2025) (shown in Table 4) indicate that adaptive weighting via Group-Relative Reward offers a consistent improvement over equal weighting, particularly in semantic alignment and overall score, confirming its potential as a robust reward fusion strategy.

Table 4: **Effect of Adaptive Reward Fusion**. We compare equal and adaptive weighting of reward signals in direct backprop. Adaptive weighting using Group Relative Reward improves overall performance.

| Method | Total | Quality | Semantic |
|---|---|---|---|
| HunyuanVideo | 83.24 | 85.09 | 75.82 |
| `PISCES`-direct (equal) | 85.05 ↑1.81 | 86.84 ↑1.75 | 77.89 ↑2.07 |
| `PISCES`-direct (adaptive) | **85.16** ↑1.92 | **86.92** ↑1.83 | **78.11** ↑2.29 |

## 4.4 OPTIMAL TRANSPORT ANALYSIS

To validate the effectiveness of OT (Villani, 2009; Cuturi, 2013) in aligning text and video embeddings, we conduct both qualitative and quantitative analyses. We extract $10,000$ text-video pairs from WebVid10M (Bain et al., 2021) using pre-trained VLM InternVideo2.

**OT Plan.** Figure 5 compares standard cross-attention maps with our token-level OT plans. While cross-attention alone produces diffuse activations, and unconstrained OT plans misalign tokens, incorporating spatio-temporal constraints in the OT cost matrix yields meaningful correspondences. This highlights the benefit of discrete OT in our Semantic Reward: it ensures fine-grained alignment of text tokens with semantically and spatio-temporally consistent video regions, directly improving localized supervision during post-training. We further validate that our designed discrete POT helps improve the video-text matching performance of pre-trained InternVideo2 by $8.11\%$ in Appendix F.

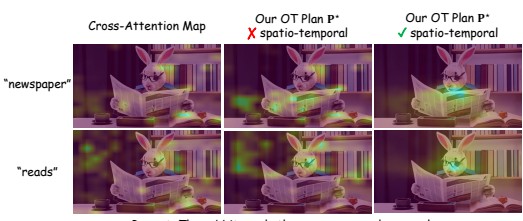

Prompt: The rabbit reads the newspaper and wears glasses.

Figure 5: Cross-attention maps (left) are diffuse, OT plan without spatio-temporal constraints (middle) misaligns tokens, while our constrained OT plan (right) produces sharper, accurate correspondences.

**Quantitative Analysis.** Mutual KNN (Huh et al., 2024) measures the alignment strength between text and video embeddings by computing the overlap in k-nearest neighbors across modalities, with higher values indicating stronger alignment. Spearman correlation coefficient $r$ evaluates structural preservation by measuring the rank similarity of embeddings before and after transformation. As shown in Table 5, compared to popular approaches, OT achieves the highest Mutual KNN and Spearman correlation, effectively aligning distributions while preserving internal structure. Post-training HunyuanVideo with OT-aligned rewards leads to the highest Quality Score and Semantic Score, validating its effectiveness in improving both overall visual quality and text-video consistency.

**OT Map.** Figure 6 (left) presents a t-SNE projection of text and video embeddings. The original text embeddings (blue) are largely misaligned with video embeddings (orange), highlighting distributional gaps in pre-trained VLMs (that lead to sub-optimal text-video alignment). OT-transformed text embeddings (green) shift significantly closer to video embeddings, demonstrating improved alignment. Figure 6 (right) further supports this observation via pairwise distance distribution analysis. The distribution of text embeddings after OT transformation closely resembles the original one, confirming OT aligns embeddings without distorting their internal relationships.

Table 5: Comparison of alignment methods. OT improves text-video alignment (higher Mutual KNN) and preserves text embedding structure (higher Spearman Correlation). Post-training with OT-aligned rewards achieves the best Quality and Semantic Scores.

| Method | Mutual KNN ↑ | Spearman Correlation ↑ | Quality Score ↑ | Semantic Score ↑ |
|---|---|---|---|---|
| Contrastive | 0.2135 | - | 85.57 | 77.00 |
| Mapping w/ L2 | 0.2318 | 0.4873 | 84.89 | 77.12 |
| Mapping w/ KL | 0.2284 | 0.4720 | 84.72 | 77.15 |
| OT (PISCES) | **0.2597** | **0.9018** | **86.73** | **80.33** |

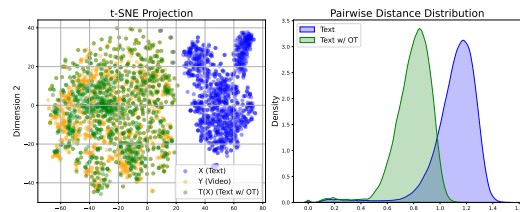

Figure 6: (Left) t-SNE shows OT aligns text embeddings (green) closer to video embeddings distribution (orange). (Right) Pairwise distance distribution indicates OT preserves text embedding structure.

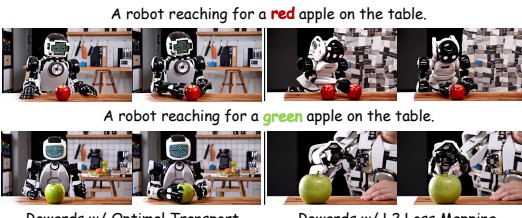

A robot reaching for a red apple on the table.

A robot reaching for a green apple on the table.

Rewards w/ Optimal Transport        Rewards w/ L2 Loss Mapping

Figure 7: Post-training with OT-aligned rewards (left) yields consistent outputs with only the expected color change, while L2 loss mapping (right) causes sampling inconsistencies, variations in appearance, and artifacts.

**Impact on T2V Post-Training.** In Figure 7, we conduct a controlled experiment using the prompt *"A robot reaching for a red/green apple on the table"* with the same random seed. Post-training with OT-aligned rewards (left) maintains structural consistency – background, robot appearance, and motion remain stable, with only the apple color changing as expected. In contrast, L2 loss mapping (right) produces unstable outputs: robot appearance and object placement vary unpredictably, and artifacts such as disappearing objects emerge. These results confirm that distorted text embeddings distribution harms reward-based post-training, leading to unstable sampling and degraded video quality. Overall, these findings underscore the advantages of OT-based alignment. By ensuring rewards operate on a well-structured space, OT prevents distributional distortions, resulting in both improved sampling stability and higher-quality T2V generation.

**Bi-objective Rewards Interaction.** To assess the interaction between OT-aligned quality and semantic rewards during training, we plot their trajectories and the cosine similarity of their gradients over time in Figure 8. Both rewards improve steadily across training steps, with final values of 0.7814 (quality) and 0.9268 (semantic) at step 256. The gradient cosine similarity stays near zero (final value 0.0074), indicating the two objectives provide orthogonal supervision and do not conflict during optimization. These findings confirm that our bi-objective formulation, combining distributional and token-level rewards, is stable and effectively supports joint improvement of visual quality and semantic alignment without trade-offs.

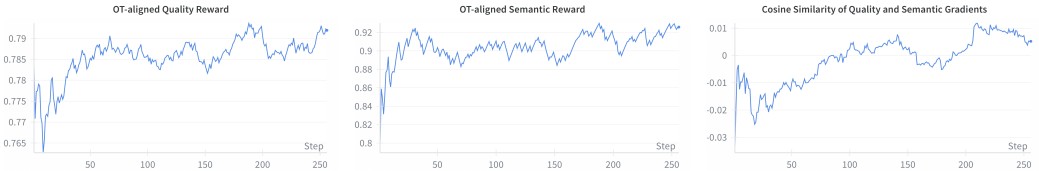

Figure 8: Both reward signals improve consistently throughout training, while their gradient cosine similarity stays close to zero, indicating stable and non-conflicting supervision from the bi-objective design.

## 5 CONCLUSION

We introduce PISCES, the first annotation-free post-training T2V algorithm that applies OT and outperforms both annotation-based and annotation-free methods on VBench and in human preference studies. Overcoming the limitation of existing annotation-free methods which rely on pre-trained VLM embeddings misaligned at both the distributional and token-levels, PISCES introduces a novel Bi-objective OT-aligned Rewards module through the lens of OT. It comprises a Distributional OT-aligned Quality Reward and a Discrete token-level Semantic Reward to significantly improve both visual quality and semantic consistency across short- and long-video generation. PISCES paves the way for scalable, principled post-training in T2V and offers a general blueprint for OT-based reward design in multimodal generation.

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

APPENDIX

In this technical appendix, we provide additional details, ablations, and analyses that support the main findings of our work. Section A introduces the Consistency Distillation (CD) mechanism used to efficiently integrate OT-aligned rewards into the `PISCES` post-training process. Section B describes the Neural Optimal Transport (NOT) formulation for aligning the distributions of text and video embeddings, while Section C presents our discrete token-level OT optimization via the entropic unbalanced Sinkhorn algorithm.

Section D demonstrates that our bi-objective reward module is compatible with different optimization paradigms (direct backpropagation and GRPO). Section E provides a detailed ablation study on the impact of OT alignment and reward types across the full 16 dimensions of VBench. Section F quantifies the impact of partial OT and structured spatio-temporal constraints on matching accuracy. Section G studies the role of motion guidance during post-training.

Section H evaluates `PISCES` under ViCLIP-based reward models to confirm generalizability across video-text encoders. Section I reports alignment performance on out-of-distribution prompts. Section J presents a hyperparameter sweep over the OT loss weights ($\gamma, \eta$) to assess sensitivity. Section K highlights a failure case of partial OT due to base encoder limitations. Section L reports inter-rater reliability and checks for category-level bias in human evaluation. Section M analyzes reward hacking risks and shows how CD loss mitigates them. Finally, Section N reports GPU-hour cost and runtime efficiency compared to baseline methods.

## A CONSISTENCY DISTILLATION

Consistency Models (CMs) (Song et al., 2023; Wang et al., 2023c) improve efficiency by enforcing self-consistency in the PF-ODE trajectory. A learned function $\boldsymbol{g} : (\mathbf{z}_t, t) \mapsto \mathbf{z}_\epsilon$ satisfies $\boldsymbol{g}(\mathbf{z}_t, t) = \boldsymbol{g}(\mathbf{z}_{t'}, t'), \quad \forall t, t' \in [\epsilon, T]$. A pre-trained diffusion model is distilled into a CM via the Consistency Distillation Loss:

$$\mathcal{L}_{\mathrm{CD}}(\boldsymbol{\theta}, \boldsymbol{\theta}^-; \phi) = \mathbb{E}_{\mathbf{z}, t}\left[d\left(\boldsymbol{g}_{\boldsymbol{\theta}}(\mathbf{z}_{t+k}, t+k), \boldsymbol{g}_{\boldsymbol{\theta}^-}(\hat{\mathbf{z}}_{t_n}^\phi, t_n)\right)\right], \tag{7}$$

where an ODE solver $\Phi$ estimates $\hat{\mathbf{z}}_{t_n}^\phi \leftarrow \mathbf{z}_{t_{n+k}} + (t_n - t_{n+k})\Phi(\mathbf{z}_{t_{n+k}}, t_{n+k}; \phi)$. To stabilize learning, EMA updates $\boldsymbol{\theta}^- \leftarrow \mathtt{stop\_grad}(\lambda\boldsymbol{\theta} + (1-\lambda)\boldsymbol{\theta}^-)$. Consistency distillation allows reward fine-tuning to backpropagate via single-step denoising, approximating multi-step denoising to enhance both efficiency and supervision signal.

## B DISTRIBUTIONAL ALIGNMENT WITH OT

We first address the embeddings distribution misalignment in pre-trained VLMs by formulating the alignment as an OT problem. Specifically, given text embeddings $\mathcal{Y}$ and real video embeddings $\mathcal{X}$ extracted from a pre-trained VLM, we learn an OT map $\mathbf{T} : \mathcal{Y} \to \mathcal{X}$ using NOT (Korotin et al., 2023). This OT mapping transforms text embeddings into a semantically-aligned space with video embeddings, significantly reducing the inherent distributional mismatch (see Figure 2 left).We define the OT problem as:

$$\sup_f \inf_{\mathbf{T}} \int_{\mathcal{X}} f(\mathbf{x}) d\nu(\mathbf{x}) + \int_{\mathcal{Y}} \left(\mathbf{c}(\mathbf{y}, \mathbf{T}(\mathbf{y})) - f(\mathbf{T}(\mathbf{y}))\right) d\mu(\mathbf{y}), \tag{8}$$

where the cost function is the squared Euclidean distance, $\mathbf{c}(\mathbf{y}, \mathbf{x}) = \|\mathbf{y} - \mathbf{x}\|^2$. We implement this via iterative optimization of the transport map $\mathbf{T}_\psi$ and potential function $f_\omega$ parameterized by neural networks, as shown in Algorithm 2. The resulting OT-aligned embeddings $\mathbf{T}^\star(\mathbf{y})$ preserve original embedding structure while aligning distributions.

## C DISCRETE TOKEN-LEVEL OT FOR SEMANTIC ALIGNMENT

**Problem setup.** Given text tokens $\{\mathbf{y}_i\}_{i=1}^N$ and video patch tokens $\{\mathbf{x}_j\}_{j=1}^M$ from a cross-attention layer, let $\mathbf{A} \in \mathbb{R}_+^{N \times M}$ denote the vanilla attention (row-normalized), $t_j$ the frame index of patch $j$,

---

**Algorithm 2** Text-Video Embeddings Distribution Alignment w/ OT

---

**Require:** text and video embedding distributions $\boldsymbol{\mu}, \boldsymbol{\nu}$; mapping network $\mathbf{T}_\psi : \mathcal{Y} \to \mathcal{X}$;
    potential network $f_\omega : \mathcal{X} \to \mathbb{R}$; number of inner iterations $K_T$; cost function $\mathbf{c} : \mathcal{Y} \times \mathcal{X} \to \mathbb{R}$
**Ensure:** learned stochastic OT map $\mathbf{T}_\psi$ representing an OT plan between distributions $\boldsymbol{\mu}, \boldsymbol{\nu}$
    **while** not converged **do**
        `unfreeze(`$\mathbf{T}_\psi$`)`; `freeze(`$f_\omega$`)`                                 $\triangleright$ **T** optimization
        **for** $k_T = 1, 2, \ldots, K_T$ **do**
            Sample a batch of text embeddings $Y \sim \boldsymbol{\mu}$
            $\mathcal{L}_{\mathbf{T}} \leftarrow \frac{1}{|Y|} \sum_{\mathbf{y} \in Y} [\mathbf{c}(\mathbf{y}, \mathbf{T}_\psi(\mathbf{y})) - f_\omega(\mathbf{T}_\psi(\mathbf{y}))]$
            Backward $\mathcal{L}_{\mathbf{T}}$ and update $\psi$ using $\frac{\partial \mathcal{L}_{\mathbf{T}}}{\partial \psi}$
        **end for**
        `freeze(`$\mathbf{T}_\psi$`)`; `unfreeze(`$f_\omega$`)`                                 $\triangleright$ $f$ optimization
        Sample batch of video and text embeddings $X \sim \boldsymbol{\nu}, Y \sim \boldsymbol{\mu}$
        $\mathcal{L}_f \leftarrow \frac{1}{|Y|} \sum_{\mathbf{y} \in \mathcal{Y}} f_\omega(\mathbf{T}_\psi(\mathbf{y})) - \frac{1}{|X|} \sum_{\mathbf{x} \in X} f_\omega(\mathbf{x})$
        Backward $\mathcal{L}_f$ and update $\omega$ using $\frac{\partial \mathcal{L}_f}{\partial \omega}$
    **end while**

---

and $s_j \in \mathbb{R}^2$ its spatial coordinate on a $h \times w$ grid. We construct a spatio-temporal, semantics-aware cost matrix $\mathbf{C} \in \mathbb{R}_+^{N \times M}$ as

$$\mathbf{C}_{ij} = \underbrace{1 - \cos(\mathbf{y}_i, \mathbf{x}_j)}_{\text{semantic}} + \gamma \underbrace{|\tau(\mathbf{y}_i) - t_j|}_{\text{temporal}} + \eta \underbrace{\|\pi(\mathbf{y}_i) - s_j\|_2}_{\text{spatial}},$$

where $\tau(\mathbf{y}_i) = \sum_k A_{ik} t_k$ and $\pi(\mathbf{y}_i) = \sum_k A_{ik} s_k$ are attention-weighted expectations of frame index and spatial position, respectively. Each component is range-normalized to $[0, 1]$, followed by a min–max normalization of $\mathbf{C}$ to $[0, 1]$ for numerical stability.

**Partial entropic OT.** Let uniform marginals $\boldsymbol{\mu} = \frac{1}{N} \mathbf{1}_N$ and $\boldsymbol{\nu} = \frac{1}{M} \mathbf{1}_M$. We solve a *partial* OT problem via an entropic, unbalanced Sinkhorn objective:

$$\min_{\mathbf{P} \geq 0} \langle \mathbf{P}, \mathbf{C} \rangle + \epsilon \sum_{i,j} \mathbf{P}_{ij} (\log \mathbf{P}_{ij} - 1) \quad \text{s.t.} \quad \mathbf{P} \mathbf{1}_M \approx \tau_a \boldsymbol{\mu}, \ \mathbf{P}^\top \mathbf{1}_N \approx \tau_b \boldsymbol{\nu},$$

where $\epsilon > 0$ is the entropic temperature and $(\tau_a, \tau_b) \in (0, 1]$ relax the marginals to achieve an effective transported fraction $m \in (0, 1]$ (we use $m = 0.9$). This yields a soft plan $\mathbf{P}^\star$ that *selectively* matches informative text tokens to consistent video regions, avoiding over-forced alignments. Algorithm 3 describes our detailed implementation of solving partial OT using entropic (unbalanced) Sinkhorn.

**Attention fusion (structure prior).** We fuse $\mathbf{P}^\star$ with vanilla attention $\mathbf{A}$ in log-space:

$$\tilde{\mathbf{A}} \propto \exp\Big( \log(\mathbf{A} + \varepsilon) + \log(\mathbf{P}^\star + \varepsilon) \Big),$$

with small $\varepsilon > 0$ for stability. Gradients flow through $\mathbf{A}$ while $\mathbf{P}^\star$ acts as a detached structural prior.

**Semantic reward.** Let `VTM` be the pre-trained video–text matching head. Using $\tilde{\mathbf{A}}$ to aggregate patch features, the Semantic Alignment Reward is

$$\mathcal{R}_{\text{OT-semantic}} = \text{softmax}\Big( \text{VTM}\big[\tilde{\mathbf{A}} \cdot \hat{\mathbf{x}}\big] \Big)_{(\text{idx}=1)}.$$

In practice, POT is applied per head and per cross-attention layer. We set $(\gamma, \eta) = (0.2, 0.2)$, $\epsilon = 0.05$, and $m = 0.9$. Please refer to Appendix F for hyperparameter selection and ablation study.

## D  AUTOMATIC EVALUATION WITH DIFFERENT OPTIMIZATION PARADIGMS

We optimize the same OT-aligned reward under two training routes: (i) direct backpropagation through the reward models and (ii) RL fine-tuning via GRPO. As reported in Table 6, both procedures yield consistent gains over the vanilla baselines on VBench (Total/Quality/Semantic) for VideoCrafter2 and HunyuanVideo, and the resulting scores are comparable across paradigms. This indicates that our reward provides meaningful supervision signals whose benefits are largely agnostic to the optimization routine.

---

**Algorithm 3** Partial OT via Entropic (Unbalanced) Sinkhorn

---

**Require:** Cost matrix $\mathbf{C} \in \mathbb{R}_+^{N \times M}$ (normalized to $[0, 1]$), entropic temperature $\epsilon > 0$, target transported fraction $m \in (0, 1]$, max iterations $K$, tolerance $\delta$
**Ensure:** Transport plan $\mathbf{P}^\star \in \mathbb{R}_+^{N \times M}$
  **Uniform marginals:** $\boldsymbol{\mu} = \frac{1}{N}\mathbf{1}_N, \ \boldsymbol{\nu} = \frac{1}{M}\mathbf{1}_M$
  **Map partial mass to unbalanced strength:**

$$\rho \leftarrow \begin{cases} \infty & \text{if } m \geq 0.999 \\ \epsilon \cdot \frac{m}{1-m} & \text{otherwise} \end{cases}, \quad \tau(\rho) \leftarrow \begin{cases} 1 & \text{if } \rho = \infty \\ \frac{\rho}{\rho+\epsilon} & \text{else} \end{cases}$$

  **Set relaxations:** $\tau_a \leftarrow \tau(\rho), \ \tau_b \leftarrow \tau(\rho)$
  **Log-kernel:** $\log \mathbf{K} \leftarrow -\mathbf{C}/\epsilon$
  Initialize $\log \mathbf{u} \leftarrow \mathbf{0}_N, \ \log \mathbf{v} \leftarrow \mathbf{0}_M; \quad \log \boldsymbol{\mu} \leftarrow \log(\boldsymbol{\mu}), \log \boldsymbol{\nu} \leftarrow \log(\boldsymbol{\nu})$
  **for** $k = 1$ to $K$ **do**
    $\log(\mathbf{Kv}) \leftarrow \log \sum_j \exp\big(\log \mathbf{K}_{:,j} + \log \mathbf{v}_j\big)$                  ▷ log-sum-exp over columns
    $\log \mathbf{u}_{\text{new}} \leftarrow \tau_a \cdot \big(\log \boldsymbol{\mu} - \log(\mathbf{Kv})\big)$
    $\log(\mathbf{K}^\top \mathbf{u}_{\text{new}}) \leftarrow \log \sum_i \exp\big(\log \mathbf{K}_{i,:} + \log \mathbf{u}_{\text{new},i}\big)$
    $\log \mathbf{v}_{\text{new}} \leftarrow \tau_b \cdot \big(\log \boldsymbol{\nu} - \log(\mathbf{K}^\top \mathbf{u}_{\text{new}})\big)$
    **if** $\max\{\|\log \mathbf{u}_{\text{new}} - \log \mathbf{u}\|_\infty, \ \|\log \mathbf{v}_{\text{new}} - \log \mathbf{v}\|_\infty\} < \delta$ **then**
      **break**
    **end if**
    $\log \mathbf{u} \leftarrow \log \mathbf{u}_{\text{new}}, \quad \log \mathbf{v} \leftarrow \log \mathbf{v}_{\text{new}}$
  **end for**
  **Recover plan:** $\log \mathbf{P} \leftarrow \log \mathbf{u}\mathbf{1}_M^\top + \log \mathbf{K} + \mathbf{1}_N \log \mathbf{v}^\top$
  $\mathbf{P}^\star \leftarrow \exp(\log \mathbf{P})$

---

Table 6: **Automatic VBench comparison on VideoCrafter2 and HunyuanVideo**. `PISCES` post-training with direct backpropagation or RL fine-tuning GRPO achieves comparable performance and shows strong improvement over the pre-trained models.

| Models | VideoCrafter2 (He et al., 2023) | | | HunyuanVideo (Kong et al., 2025) | | |
|---|---|---|---|---|---|---|
| | Total | Quality | Semantic | Total | Quality | Semantic |
| Vanilla | 80.44 | 82.20 | 73.42 | 83.24 | 85.09 | 75.82 |
| PISCES (Direct Backpropagation) | 82.51 ↑2.07 | 83.73 ↑1.53 | **77.63** ↑4.21 | 85.05 ↑1.81 | **86.84** ↑1.75 | 77.89 ↑2.07 |
| PISCES (GRPO) | **82.75** ↑2.31 | **84.05** ↑1.85 | 77.54 ↑4.12 | **85.45** ↑2.21 | 86.73 ↑1.64 | **80.33** ↑4.51 |

## E    COMPREHENSIVE ABLATION STUDY

**Effectiveness of OT Alignment.** Comparing `PISCES` w/o OT (which post-trains with pre-trained VLM embeddings) against full `PISCES` in Table 7, we observe significant improvements in both Quality Score ($83.44 \rightarrow 83.73$) and Semantic Score ($75.82 \rightarrow 77.63$). This confirms that aligning text-video distributions before post-training enhances both global coherence and fine-grained text-video correspondence.

**Impact of Quality and Semantic Alignment Rewards.** To assess the individual effects of OT-aligned Quality and Semantic Reward, we compare `PISCES` w/ $\mathcal{R}_{\text{OT-quality}}$ and $\mathcal{R}_{\text{OT-semantic}}$ separately. Quality Reward primarily improves global coherence, reflected in gains in Quality Score (83.77), Aesthetic Quality (66.92), and Subject Consistency (97.07). Meanwhile, Semantic Reward enhances fine-grained alignment, leading to improvements in Semantic Score (76.99), Human Action (96.60), and Spatial Relation (44.97).

**Full `PISCES`.** Integrating both rewards (full `PISCES`) results in the highest Total Score (82.51). Notably, Overall Consistency (29.10) and Temporal Style (26.97) also improve, reinforcing that the combination of OT alignment and both rewards provides the best optimization signal for text-video post-training.

## F    OPTIMAL TRANSPORT PLAN ANALYSIS

Table 8 analyzes the effect of Partial OT and spatio-temporal constraints on video-text matching within InternVideo2. We evaluate on 10,000 video-text pairs sampled from WebVid10M (Bain et al., 2021).

Table 7: Ablation Study. We analyze the contributions of OT alignment and OT-aligned Rewards to post-training performance. PISCES w/o OT post-trains with pre-trained VLM embeddings, while PISCES w/ $\mathcal{R}_{\text{OT-semantic}}$ and PISCES with $\mathcal{R}_{\text{OT-semantic}}$ assess the impact of OT-aligned Quality and Semantic Rewards, respectively. Full PISCES, which integrates OT alignment and both rewards, achieves the best performance across Total, Quality, and Semantic Scores, demonstrating the effectiveness of structured reward optimization. Bold numbers denote the best results in each category.

| Method | OT | $\mathcal{R}_{\text{semantic}}$ | $\mathcal{R}_{\text{quality}}$ | Total Score | Quality Score | Subject Consist. | BG Consist. | Temporal Flicker. | Motion Smooth. | Aesthetic Quality | Dynamic Degree | Image Quality |
|---|---|---|---|---|---|---|---|---|---|---|---|---|
| VideoCrafter2 | ✗ | ✗ | ✗ | 80.44 | 82.20 | 96.85 | 98.22 | 98.41 | 97.73 | 63.13 | 42.50 | 67.22 |
| PISCES w/o OT | ✗ | ✓ | ✓ | 81.92 | 83.44 | 96.99 | 97.66 | 98.02 | 97.16 | 66.39 | 52.78 | 70.50 |
| PISCES w/ $\mathcal{R}_{\text{OT-quality}}$ | ✓ | ✗ | ✓ | 82.21 | **83.77** | 97.07 | 97.58 | 97.72 | 97.10 | **66.92** | **58.06** | 70.56 |
| PISCES w/ $\mathcal{R}_{\text{OT-semantic}}$ | ✓ | ✓ | ✗ | 81.70 | 82.87 | **97.59** | **98.61** | 98.06 | **97.31** | 66.83 | 40.00 | 70.19 |
| PISCES | ✓ | ✓ | ✓ | **82.51** | 83.73 | 96.61 | 97.49 | **98.72** | 96.80 | 66.07 | 57.50 | 70.39 |

| Method | OT | $\mathcal{R}_{\text{semantic}}$ | $\mathcal{R}_{\text{quality}}$ | Semantic Score | Object Class | Multiple Objects | Human Action | Color | Spatial Relation. | Scene | Appear. Style | Temporal Style | Overall Consist. |
|---|---|---|---|---|---|---|---|---|---|---|---|---|---|
| VideoCrafter2 | ✗ | ✗ | ✗ | 73.42 | 92.55 | 40.66 | 95.00 | **92.92** | 35.86 | 55.29 | **25.13** | 25.84 | 28.23 |
| PISCES w/o OT | ✗ | ✓ | ✓ | 75.82 | 95.57 | 53.06 | 97.80 | 90.52 | 39.51 | 59.07 | 24.27 | 26.03 | 28.26 |
| PISCES w/ $\mathcal{R}_{\text{OT-quality}}$ | ✓ | ✗ | ✓ | 75.97 | 94.84 | 57.80 | **98.00** | 90.36 | 38.51 | 55.54 | 24.45 | 26.37 | 28.62 |
| PISCES w/ $\mathcal{R}_{\text{OT-semantic}}$ | ✓ | ✓ | ✗ | 76.99 | 95.32 | 59.36 | 96.60 | 91.06 | **44.97** | **59.93** | 24.04 | 25.81 | 28.26 |
| PISCES | ✓ | ✓ | ✓ | **77.63** | **98.13** | **66.51** | 97.60 | 92.46 | 36.07 | 58.75 | 23.53 | **26.97** | **29.10** |

Table 8: **Ablation study on Partial OT and spatio-temporal constraints.** We report Video-Text Matching (VTM) accuracy on 10,000 video-text pairs from WebVid10M. Partial OT with $m = 0.9$ achieves the best alignment, while our spatio-temporal constraints ($\gamma = 0.2$, $\eta = 0.2$) further boost performance. Arrows indicate relative change compared to vanilla cross-attention baseline.

| Method | VTM Acc. (%) $\uparrow$ | Change |
|---|---|---|
| Vanilla cross-attention | 81.25 | – |
| Full OT ($m = 1.0$) | 86.87 | ↑5.62 |
| Partial OT ($m = 0.5$) | 78.92 | ↓2.33 |
| Partial OT ($m = 0.9$) | 87.54 | ↑6.29 |
| Partial OT ($m = 0.9$) + spatial only ($\eta = 0.2$) | 88.17 | ↑6.92 |
| Partial OT ($m = 0.9$) + temporal only ($\gamma = 0.2$) | 87.98 | ↑6.73 |
| Partial OT ($m = 0.9$) + spatio-temporal ($\gamma = \eta = 0.1$) | 87.06 | ↑5.81 |
| Partial OT ($m = 0.9$) + spatio-temporal ($\gamma = \eta = 0.2$) | **89.36** | **↑8.11** |

We observe that using Partial OT with a mass parameter $m = 0.9$ achieves the best score of **89.36%**, improving by **+8.11%** over vanilla cross-attention. This indicates that not all text tokens need to be matched to visual patches. For example, uninformative words (*e.g.*, articles or stopwords) need not be explicitly grounded in the visual domain. Allowing partial transport filters out such noisy matches while preserving key semantic correspondences. Conversely, setting $m = 0.5$ removes too many tokens, causing essential words to be ignored and leading to degraded alignment.

Regarding constraints, our designed cost function with both spatial ($\eta = 0.2$) and temporal ($\gamma = 0.2$) penalties yields the highest performance, boosting video-text matching by **8.11%** (from 81.25% to 89.36%). This demonstrates that integrating spatio-temporal structure into OT provides sharper and more accurate token-level correspondences, thereby enhancing fine-grained text-video alignment. Overall, these results confirm the effectiveness of Partial OT with structured constraints in improving alignment quality.

## G   MOTION GUIDANCE IN T2V POST-TRAINING

For a fair comparison to highlight the impact of rewards, we provide addition ablations (see Table 9) where we evaluate both methods with and without motion guidance. Without motion guidance, T2V-Turbo-v2 (Li et al., 2025) performance drops to 83.26 (Quality) and 76.30 (Semantic) on VideoCrafter2 (He et al., 2023). In comparison, PISCES-direct still achieves stronger quality (83.73, +0.47) and significantly higher semantic alignment (77.63, +1.33), highlighting the effectiveness of our OT-aligned rewards. Conversely, by adding motion guidance to PISCES-direct makes it outperform

Table 9: **Effect of Motion Guidance on post-training VideoCrafter2 and HunyuanVideo**. PISCES significantly outperforms T2V-Turbo-v2 (Li et al., 2025) in both with and without motion guidance settings.

| Models | VideoCrafter2 (He et al., 2023) | | | HunyuanVideo (Kong et al., 2025) | | |
|---|---|---|---|---|---|---|
| | Total | Quality | Semantic | Total | Quality | Semantic |
| Vanilla | 80.44 | 82.20 | 73.42 | 83.24 | 85.09 | 75.82 |
| T2V-Turbo-v2 *w/o motion* (Li et al., 2025) | 81.87 ↑1.43 | 83.26 ↑1.06 | 76.30 ↑2.88 | 84.25 ↑1.01 | 85.93 ↑0.84 | 77.52 ↑1.70 |
| PISCES (Direct Backpropagation) w/o motion | **82.51** ↑2.07 | **83.73** ↑1.53 | **77.63** ↑4.21 | **85.05** ↑1.81 | **86.84** ↑1.75 | **77.89** ↑2.07 |
| T2V-Turbo-v2 *w/ motion* (Li et al., 2025) | 82.34 ↑1.90 | 83.93 ↑1.73 | 75.97 ↑2.55 | 84.50 ↑1.26 | 86.32 ↑1.23 | 77.24 ↑1.42 |
| PISCES (Direct Backpropagation) w/ motion | **82.79** ↑2.35 | **84.12** ↑1.92 | **77.45** ↑4.03 | **85.24** ↑2.00 | **87.07** ↑1.98 | **77.94** ↑2.12 |

Table 10: **Comparison of Evaluators for PISCES on VBench**. Both InternVideo2 and ViCLIP as evaluators yield strong improvements over the Vanilla baseline, confirming PISCES' robustness across feature extractors.

| Models | VideoCrafter2 (He et al., 2023) | | | HunyuanVideo (Kong et al., 2025) | | |
|---|---|---|---|---|---|---|
| | Total | Quality | Semantic | Total | Quality | Semantic |
| Vanilla | 80.44 | 82.20 | 73.42 | 83.24 | 85.09 | 75.82 |
| PISCES w/ InternVideo2 (Wang et al., 2024) | 82.75 ↑2.31 | 84.05 ↑1.85 | 77.54 ↑4.12 | 85.45 ↑2.21 | 86.73 ↑1.64 | 80.33 ↑4.51 |
| PISCES w/ ViCLIP (Wang et al., 2023b) | 82.84 ↑2.40 | 84.18 ↑1.98 | 77.49 ↑4.07 | 85.33 ↑2.09 | 86.77 ↑1.68 | 79.58 ↑3.76 |

T2V-Turbo-v2 across all metrics on VideoCrafter2. More notably, on HunyuanVideo (Kong et al., 2025), PISCES-direct *without motion guidance* already **outperforms T2V-Turbo-v2 with motion** in all metrics (85.05 vs 84.50 Total Score), and this advantage is also reflected in our human preference study (Figure 3). This clearly demonstrates that our reward formulation–rather than the optimization strategy or motion module–is the key driver behind the improvements.

## H PISCES WITH VICLIP EVALUATOR

To test whether PISCES improvements generalize beyond InternVideo2 (Wang et al., 2024), we conducted a controlled experiment where both our rewards (Distributional OT and Semantic OT) are based on ViCLIP (Wang et al., 2023b), a CLIP-based video–text encoder that differs from InternVideo2 in training corpus and representation space. The results in Table 10 confirm that PISCES remains effective even when rewards and evaluation use ViCLIP, achieving comparable performance across both benchmarks and further mitigating concerns of reward overfitting to a specific model family. These consistent results show the generality of our OT-aligned reward formulation and demonstrate that the observed improvements are not confined to InternVideo2 features.

## I SCOPE OF GENERALIZATION

To assess the robustness of PISCES beyond the WebVid10M (Bain et al., 2021) and VidGen-1M (Tan et al., 2024) domain, we curated 100 diverse out-of-distribution (OOD) prompts. These prompts cover challenging and underrepresented scenarios including robotics actions, embodied tasks, procedural instructions, abstract concepts, and rare object-event compositions. Example prompts include:

- *A robot arm with a red gripper picks up a blue cube and sorts it into a green bin on a moving conveyor belt in a bright factory hall.*

- *A tiny mechanical mouse navigates through a labyrinth of gears inside a giant clock, camera close-up on its delicate paws.*

To test alignment under these harder OOD conditions, we compare the cosine similarity between text prompt embeddings and generated video embeddings using the ViCLIP (Wang et al., 2023b) encoder. Results in Table 11 confirm that PISCES exhibits stronger alignment even under distribution shifts and when evaluated using an independent video-text encoder, *e.g.* ViCLIP (Wang et al., 2023b), supporting the generality and robustness of our OT-aligned rewards.

Table 11: **OOD Alignment Performance using ViCLIP (Wang et al., 2023b).** `PISCES` achieves the highest cosine similarity, confirming robustness under distribution shifts.

| Method | Cosine Similarity ↑ |
|---|---|
| HunyuanVideo (Kong et al., 2025) | 0.4128 |
| T2V-Turbo-v2 (Li et al., 2024a) | 0.4390 |
| VideoReward-DPO (Liu et al., 2025a) | 0.4285 |
| PISCES | **0.4517** |

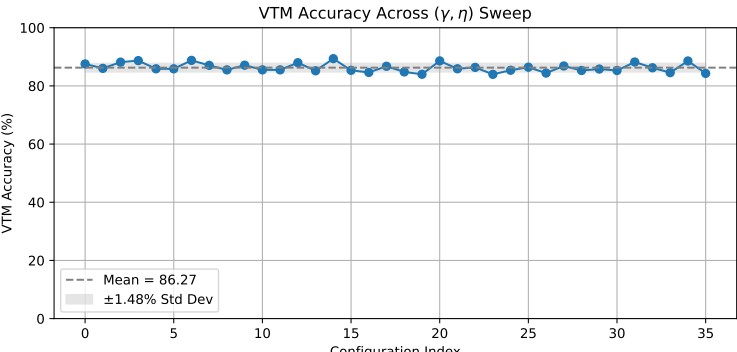

Figure 9: VTM accuracy under $(\gamma, \eta)$ sweep. Despite configuration variations, the accuracy remains stable ($\pm 1.48\%$ std).

## J HYPERPARAMETERS SENSITIVITY ANALYSIS

To further assess robustness, we conduct a stability analysis by sweeping $(\gamma, \eta)$ over the range $[0.0, 0.5]$ in 0.1 increments. For each setting, we measure the Video-Text Matching (VTM) accuracy on 10,000 WebVid10M (Bain et al., 2021) video-text pairs using our OT-aligned Semantic Reward with InternVideo2 (Wang et al., 2024).

We observe that the VTM accuracy varies smoothly across this range, with a standard deviation of only $1.48\%$, indicating that our method is robust to these weights. This confirms the stability of our Partial OT formulation with respect to its structured penalty terms. As shown in Figure 9, the variation is minimal and has negligible impact on performance.

## K FAILURE EXAMPLE OF DISCRETE PARTIAL OT

We provide a qualitative example in Figure 10 that illustrates both the sensitivity and limitations of OT alignment. **Left:** The attention map from vanilla cross-attention fails to ground the token *"glasses"*. **Middle:** Our OT plan $\mathbf{P}^\star$ with partial mass $m = 0.5$ suppresses noise but also removes the valid token *"glasses"*, reproducing misalignment. **Right:** With mass $m = 0.9$, the OT plan retains the *"glasses"* token and aligns it better–but still imperfectly, activating only the right lens.

This reveals a current limitation of the setup: while the OT module improves semantic alignment through structured grounding, overall performance depends on the representational precision of the underlying video–text model, InternVideo2 (Wang et al., 2024), which was not optimized for fine-grained localization tasks such as segmentation. As a result, even a well-structured transport plan may not fully resolve detailed grounding errors when the base features do not capture sufficient spatial detail. Our approach remains agnostic to the underlying base model, and continued progress in open-source VLMs is likely to further enhance fine-grained grounding performance.

## L HUMAN PREFERENCE DETAILS

To assess inter-rater reliability, we computed Fleiss' Kappa over the collected human preference judgments. Across the three evaluation axes (visual quality, motion quality, and semantic alignment), we obtained an average Fleiss' Kappa score of 0.72, indicating substantial agreement among raters.

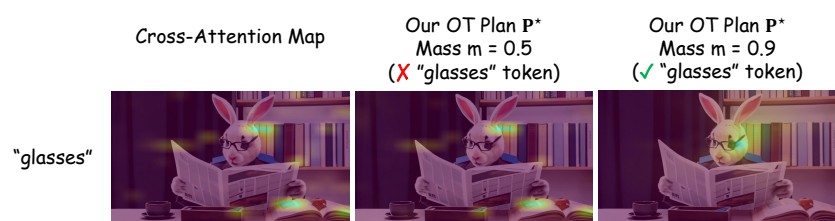

Prompt: The rabbit reads the newspaper and wears glasses.

Figure 10: **Effect of OT mass parameter.** Low OT mass ($m = 0.5$) fails to retain the *"glasses"* token, while higher mass ($m = 0.9$) improves alignment.

Table 12: **Impact of CD Loss on Reward Stability and Quality.** CD loss serves as a regularizer that prevents overfitting to reward signals and stabilizes training.

| Method | Total Score | Quality | Semantic |
|---|---|---|---|
| HunyuanVideo (Kong et al., 2025) | 83.24 | 85.09 | 75.82 |
| PISCES w/o CD Loss | 84.80 ↑1.56 | 86.51 ↑1.42 | 77.95 ↑2.13 |
| PISCES w/ CD Loss | 85.05 ↑1.81 | 86.84 ↑1.75 | 77.89 ↑2.07 |

Participants were aged between 18 and 38, with diverse backgrounds including art, engineering, and computer science. All raters were blinded to method identity to prevent bias.

We analyzed category-level bias by classifying 400 prompts into 290 motion-heavy (e.g., running, jumping) and 110 static (e.g., portraits, scenic shots). The Pearson correlation between category type and preference for our method is 0.038, indicating no significant bias–our method performs consistently across both motion-heavy and static prompts.

## M    REWARD HACKING AND MITIGATION STRATEGY

We found that the Consistency Distillation (CD) loss helps mitigate reward hacking and stabilize training. Intuitively, this loss anchors the student model (updated via reward supervision) to the teacher model's original distribution, serving both as a regularizer and as an efficient training mechanism by enabling fewer denoising steps.

We evaluate the impact of CD loss in Table 12. Without CD loss, the semantic reward is optimized more aggressively, leading to a noticeable drop in visual quality (from 86.84 to 86.51) and total score (from 85.05 to 84.80), confirming that the model may over-optimize the reward signal at the expense of generation fidelity. By retaining CD loss, we balance the original model performance and reward-driven improvements.

## N    TRAINING COST AND EFFICIENCY ANALYSIS

To train the Neural OT (NOT) map, we use video-text pairs with 8-frame clips, extracted using frozen InternVideo2, and train on a single A100 GPU for one day, equivalent to 24 A100 GPU-hours. In comparison, annotation-based pipelines such as VideoReward-DPO require 72 A800 GPU-hours to train their reward models.

The total wall-clock cost of PISCES post-training is 29.78 GPU-hours on 8×A100s, slightly higher than the 26.52 GPU-hours required by T2V-Turbo-v2 (training time only, without evaluation). While PISCES incurs a marginal increase in training time, this is a one-time cost and remains negligible compared to the massive pre-training costs of T2V models-665,000 GPU-hours for Seaweed-7B and even more for HunyuanVideo-13B. Thus, PISCES provides a lightweight and effective reward alignment mechanism with minimal computational overhead.

Finally, inference incurs no additional cost from the reward models. In fact, for GRPO-tuned models, we reduce the denoising steps from 50 to 16 through consistency distillation, resulting in approximately 3× faster inference.

