# OpenReview forum: "PISCES: Annotation-free Text-to-Video Post-Training via Bi-objective OT-aligned Rewards"
_ICLR.cc/2026/Conference — ICLR 2026 Conference Desk Rejected Submission_

### Official Review · Reviewer_5pWq · 2025-10-27

**Soundness:** 3
**Presentation:** 3
**Contribution:** 3
**Rating:** 6
**Confidence:** 4

**Summary:**

PISCES is an annotation-free post-training method that aligns reward signals with human judgment via a Bi-objective OT-aligned Rewards module. It uses optimal transport to bridge text–video embeddings at two levels:
(1) a Distributional OT-aligned Quality Reward capturing overall visual quality and temporal coherence, and (2) a Discrete token-level OT-aligned Semantic Reward enforcing semantic and spatio-temporal correspondence between text tokens and video patches. This dual-objective design provides rich, label-free supervision that improves text–video alignment and fidelity during post-training.

**Strengths:**

1. Annotation-free and scalable: Post-training without human labels reduces cost and bias, enabling easy scaling to large datasets.

2. Bi-objective rewards: Combines a distributional OT-aligned quality reward with a token-level OT-aligned semantic reward.

3. Robustness to noise/misalignment: Spatio-temporal constraints and partial OT selectively match salient tokens/patches, suppressing spurious alignments.

**Weaknesses:**

1. Insufficient differentiation from prior OT/POT work.
The paper does not clearly articulate what is fundamentally new relative to existing optimal-transport or partial-OT approaches for cross-modal alignment and reward shaping.

2. Limited analysis of bi-objective interactions.
The interaction between the quality-level and token-level rewards is not characterized. There are no training/reward curves or gradient cosine-similarity analyses to diagnose potential conflicts, leaving stability and robustness unclear.

3. Risk of reward hacking without mitigation evidence.
The method may optimize one objective at the expense of the other, but the paper presents fewer empirical checks. Mitigation strategies and failure case studies are necessary to assess robustness.

**Questions:**

1. What is fundamentally new relative to existing OT/partial-OT methods for cross-modal alignment and reward shaping? Please specify the conceptual and technical differences

2. Do the two objectives interact adversely during training? Please report training/reward trajectories and the cosine similarity between their gradients, and discuss any observed conflicts.

3. Have you observed reward hacking (e.g., improving one objective degrades the other or overall performance)? If so, what mitigation strategies did you employ, and how effective were they?

---

> ### Author Response · Authors · 2025-11-23
> **Response to Reviewer 5pWq (1/2)**
>
> We thank the reviewer for your thoughtful feedback. We address your comments below and have incorporated the feedback (main paper Section 3.2, 4.4, Appendix M).
>
>
> >**Q1: Insufficient differentiation from prior OT/POT work. The paper does not clearly articulate what is fundamentally new relative to existing optimal-transport or partial-OT approaches for cross-modal alignment and reward shaping. What is fundamentally new relative to existing OT/partial-OT methods for cross-modal alignment and reward shaping? Please specify the conceptual and technical differences.**
>
> **Conceptual Novelty**
>
> Most prior OT/POT work focuses on cross‑modal retrieval, temporal grounding, or representation learning. None has explored OT for reward modeling in post‑training text-to-video (T2V) generative models.
>
> PISCES is, to our knowledge, the first to:
> - Identify that pre‑trained VLMs can produce misaligned text–video embeddings, which undermines annotation‑free reward supervision for post-training text-to-video models. (mentioned in first contribution)
> - Reframe OT from an alignment/retrieval tool into a reward‑shaping mechanism that directly supervises text-to-video generative models. (mentioned in Related Work)
> - Design bi-objective rewards that reflects how humans judge videos: (1) distribution-level for overall quality and (2) token-level for semantic alignment. (mentioned in Related Work)
>
>
> **Technical Novelty**
> - We introduce a structured Partial OT formulation specifically designed for T2V post-training with the cost matrix: semantic similarity, temporal and spatial constraints. This yields a transport plan $\mathbf{P}^\star$ that selectively grounds meaningful tokens while ignoring uninformative ones. (We have highlighted this more in the Method Section.)
>
> - Instead of modifying the VLM, we propose to inject OT structure via log-space fusion: $\tilde{\mathbf{A}} \propto \exp \big(\log(\mathbf{A} + \varepsilon) + \log(\mathbf{P}^\star + \varepsilon)\big)$, a lightweight, differentiable mechanism that cleanly integrates POT guidance into all cross‑attention layers. (We have highlighted this more in the Method Section.)
>
> - As shown in Figure 5 and Table 8 (included in original submission, just changed the order), our spatio‑temporal POT formulation substantially outperforms vanilla POT or cross‑attention baselines, verifying that our design provides new, effective OT‑based reward shaping.
>
> ---
>
> >**Q2: Limited analysis of bi-objective interactions. The interaction between the quality-level and token-level rewards is not characterized. There are no training/reward curves or gradient cosine-similarity analyses to diagnose potential conflicts, leaving stability and robustness unclear. Do the two objectives interact adversely during training? Please report training/reward trajectories and the cosine similarity between their gradients, and discuss any observed conflicts.**
>
> To assess the interaction between our quality and semantic rewards, we have added both reward trajectories and the cosine similarity of their gradients during post-training. As shown in Figure 8 in Sec 4.4 main paper, both rewards steadily improve, confirming they contribute constructively. The gradient cosine similarity fluctuates around zero and settles slightly above zero at the end of training, indicating that the two objectives are orthogonal and do not conflict during optimization. These results demonstrate the stability of our bi-objective design and its effectiveness in  enhancing both quality and semantic alignment in video generation.

---

> > ### Author Response · Authors · 2025-11-23
> > **Response to Reviewer 5pWq (2/2)**
> >
> > >**Q3: Risk of reward hacking without mitigation evidence. The method may optimize one objective at the expense of the other, but the paper presents fewer empirical checks. Mitigation strategies and failure case studies are necessary to assess robustness. Have you observed reward hacking (e.g., improving one objective degrades the other or overall performance)? If so, what mitigation strategies did you employ, and how effective were they?**
> >
> > We found that CD loss helps mitigate reward hacking and stabilize training. Intuitively, this loss anchors the student model (updated via reward supervision) to the original teacher model’s distribution, serving both as a regularizer and as an efficient training mechanism (fewer denoising steps).
> >
> > We evaluate the impact of CD loss in the table below. Without CD loss, the semantic reward is optimized more aggressively which leads to a noticeable drop in visual quality (from 86.84 to 86.51) and total score (from 85.05 to 84.80), confirming that the model may over-optimize the reward signal at the expense of generation fidelity (i.e., reward hacking). By retaining CD loss, we balance the original model performance and reward-driven improvements.
> >
> >
> > ### Impact of CD Loss on Reward Stability and Quality
> >
> > | Method              | Total Score ↑     | Quality ↑         | Semantic ↑        |
> > |----------------------|-------------------|--------------------|--------------------|
> > | HunyuanVideo         | 83.24              | 85.09              | 75.82              |
> > | PISCES w/o CD Loss   | 84.80 (↑1.56)      | 86.51 (↑1.42)      | 77.95 (↑2.13)      |
> > | PISCES w/ CD Loss    | 85.05 (↑1.81)      | 86.84 (↑1.75)      | 77.89 (↑2.07)      |
> >
> > These findings highlight CD loss as a simple yet effective mitigation strategy to reduce reward hacking and ensure robust and efficient post-training behavior.
> >
> > We have added this analysis on reward hacking mitigation in Appendix M.

---

> ### Comment · Reviewer_5pWq · 2025-11-28
>
> I appreciate the additional experiments conducted for the rebuttal and detailed analysis, which have adequately addressed my concerns.
>
> For the reasons mentioned above, I recommend accepting this work.

---

### Official Review · Reviewer_2exC · 2025-10-28

**Soundness:** 3
**Presentation:** 3
**Contribution:** 3
**Rating:** 6
**Confidence:** 4

**Summary:**

The paper presents PISCES, an annotation-free post-training framework for text-to-video (T2V) that replaces vanilla VLM-based reward signals with a Bi-objective OT-aligned Rewards module. The key idea is to align the reward embedding space to better reflect human judgments, addressing that pre-trained VLM embeddings are misaligned across text/video modalities. Concretely: (i) a Distributional OT-aligned Quality Reward learns a neural OT map (NOT) T⋆ that projects text embeddings onto the real-video embedding manifold; quality is computed as cosine similarity of [CLS] representations T⋆(y[CLS]) and x̂[CLS]. (ii) a Discrete token-level OT-aligned Semantic Reward computes a partial OT plan P⋆ over spatio-temporal cost to align text tokens to relevant video patches and feeds the fused attention map to a VTM head to produce a semantic reward. The module supervises fine-tuning either via direct backprop through reward models or via RL (GRPO). On VBench, PISCES improves both Quality and Semantic scores over annotation-free baselines (T2V-Turbo, -v2) and annotation-based methods (VideoReward-DPO, UnifiedReward), for both short-video (VideoCrafter2) and long-video (HunyuanVideo). Human preferences also favor PISCES in visual quality, motion quality, and alignment.

**Strengths:**

- Clear problem diagnosis: annotation-free reward methods depend on misaligned text/video embeddings; the paper substantiates this with Mutual KNN and Spearman rank results and t-SNE visualizations.
- Principled remedy using OT: (a) a learned distributional OT map T⋆ to align text embeddings to the real-video manifold; (b) a partial OT plan with spatio-temporal costs to enforce fine-grained token grounding. Both are well-motivated and integrated cleanly into post-training.
- Strong empirical evidence: consistent gains on VBench over both annotation-free (T2V-Turbo, v2) and annotation-based (VideoReward-DPO, UnifiedReward) post-training; results hold across two generators (VideoCrafter2/HunyuanVideo) and with both direct backprop and GRPO.
- Ablations are informative: OT vs. no-OT; quality vs. semantic reward; partial OT mass m; spatial/temporal penalties; analysis of stability under consistent seeds.

**Weaknesses:**

- Metric alignment risk: Rewards use InternVideo2 features; VBench and human studies show gains, but further measurement with independent evaluators (e.g., alternative video-text encoders or third-party human studies) would mitigate concerns of “reward overfitting” to the same representation family.
- Compute/efficiency: Learning NOT (OT map) and solving discrete POT per layer/head adds cost. Report training-time overhead and inference-time impact (if any) and compare to annotation-based pipelines.
- Scope of generalization: Results are on WebVid10M/VidGen-1M distribution. Evaluate on distinct domains (e.g., robotics prompts with compositional actions) and under harder OOD shifts to stress-test the alignment benefits.
- Design sensitivity: The spatio-temporal cost is hand-weighted (γ, η). More extensive sensitivity analysis or a learned weighting strategy could make it more robust.

**Questions:**

- Evaluator diversity: Can you add evaluations using a different video–text model (e.g., CLIP-based video encoders) to demonstrate that improvements are not specific to InternVideo2 features?
- Cost accounting: What is the wall-clock/GPU-day overhead of NOT and POT modules during post-training? Any measurable inference-time overhead when using GRPO-tuned models?
- Failure modes: Show qualitative failure cases where OT alignment harms outputs (e.g., biased mappings or spurious correlations). Any cases where partial OT mass m or constraints degrade alignment?
- Human preference details: Provide inter-rater agreement and demographic diversity; were raters blind to method identity? Any observed category-level biases (e.g., motion-heavy vs. static prompts)?
- Reward fusion: For direct backprop, how do you weight the rewards vs. the consistency loss? Any benefit to adaptive weighting guided by variance or confidence in reward estimates?

---

> ### Author Response · Authors · 2025-11-23
> **Response to Reviewer 2exC (1/3)**
>
> We thank the reviewer for your thoughtful feedback. We address your comments below and have incorporated the feedback (main paper Section 4.3, Appendix H, I, J, K, L, N).
>
> >**Q1: Metric alignment risk: Rewards use InternVideo2 features; VBench and human studies show gains, but further measurement with independent evaluators (e.g., alternative video-text encoders or third-party human studies) would mitigate concerns of “reward overfitting” to the same representation family. Evaluator diversity: Can you add evaluations using a different video–text model (e.g., CLIP-based video encoders) to demonstrate that improvements are not specific to InternVideo2 features?**
>
> To test whether PISCES improvements generalize beyond InternVideo2, we conducted a controlled experiment where both our rewards (Distributional OT and Semantic OT) are based on ViCLIP, a CLIP-based video–text encoder that differs from InternVideo2 in training corpus and representation space.
>
> The results below confirm that PISCES remains effective even when rewards and evaluation use ViCLIP, achieving comparable performance across both benchmarks and further mitigating concerns of reward overfitting to a specific model family.
>
>
> ### VBench Results with InternVideo2 vs. ViCLIP Evaluators
>
> | Model                  | Total (VC2)      | Quality (VC2)     | Semantic (VC2)    | Total (HY)       | Quality (HY)      | Semantic (HY)     |
> |------------------------|------------------|--------------------|--------------------|-------------------|--------------------|--------------------|
> | Vanilla                | 80.44            | 82.20              | 73.42              | 83.24             | 85.09              | 75.82              |
> | PISCES (InternVideo2)  | 82.75 (↑2.31)  | 84.05 (↑1.85)    | 77.54 (↑4.12)    | 85.45 (↑2.21)   | 86.73 (↑1.64)    | 80.33 (↑4.51)    |
> | PISCES (ViCLIP)        | 82.84 (↑2.40)  | 84.18 (↑1.98)    | 77.49 (↑4.07)    | 85.33 (↑2.09)   | 86.77 (↑1.68)    | 79.58 (↑3.76)    |
>
> These consistent results show the generality of our OT-aligned reward formulation and demonstrate that the observed improvements are not confined to InternVideo2 features. We have added this experiment with ViCLIP evaluator in Appendix H.
>
> We conducted third-party human studies prior to the main paper submission, and the results are presented in Figure 3.
>
> ---
> >**Q2: Compute/efficiency: Learning NOT (OT map) and solving discrete POT per layer/head adds cost. Report training-time overhead and inference-time impact (if any) and compare to annotation-based pipelines. Cost accounting: What is the wall-clock/GPU-day overhead of NOT and POT modules during post-training? Any measurable inference-time overhead when using GRPO-tuned models?**
>
>
> To train the Neural OT (NOT) map, we use video-text pairs with 8-frame clips, extracted using frozen InternVideo2, and train on a single A100 GPU for one day, equivalent to 24 A100 GPU-hours (training only, without evaluation during training). In comparison, annotation-based pipelines like VideoReward-DPO require 72 A800 GPU-hours to train their reward models. We have added the NOT training in main paper, implementation details.
>
> The total wall-clock cost of PISCES post-training is 29.78 GPU-hours on 8×A100s, fractionally high compared to 26.52 GPU-hours taken by T2V-Turbo-v2 (training time only, without evaluation during training). While the overall training cost is slightly higher than the post-training T2V baseline, this increase is a one-time overhead and remains negligible when compared to the massive pre-training cost of T2V models-665,000 GPU-hours for Seaweed-7B and even more for HunyuanVideo-13B. Thus, PISCES provides a lightweight and effective reward alignment mechanism with minimal additional computational overhead.
>
> Finally, inference incurs no additional cost from reward models. In fact, for GRPO-tuned models, we reduce the denoising steps from 50 to 16 due to consistency distillation, resulting in ~3× faster inference. We have included these analysis in Appendix N.

---

> ### Author Response · Authors · 2025-11-23
> **Response to Reviewer 2exC (2/3)**
>
> >**Q3: Scope of generalization: Results are on WebVid10M/VidGen-1M distribution. Evaluate on distinct domains (e.g., robotics prompts with compositional actions) and under harder OOD shifts to stress-test the alignment benefits.**
>
> To assess the robustness of PISCES beyond the WebVid10M/VidGen-1M domain, we curated 100 diverse OOD prompts. These prompts cover challenging and underrepresented scenarios including robotics actions, embodied tasks, procedural instructions, abstract concepts, and rare object-event compositions. Example prompts include:
> - “A robot arm with a red gripper picks up a blue cube and sorts it into a green bin on a moving conveyor belt in a bright factory hall.”
> - “A tiny mechanical mouse navigates through a labyrinth of gears inside a giant clock, camera close-up on its delicate paws.”
>
> To test alignment under these harder OOD conditions, we compare the cosine similarity between text prompt embeddings and generated video embeddings using the ViCLIP encoder. Results are reported below:
>
> | Model                   | Cosine Similarity ↑ |
> |-------------------------|---------------------|
> | HunyuanVideo            | 0.4128              |
> | T2V-Turbo-v2            | 0.4390              |
> | VideoReward-DPO         | 0.4285              |
> | PISCES                  | **0.4517**          |
>
> These results confirm that PISCES exhibits stronger alignment even under distribution shifts and when evaluated using an independent video-text encoder (ViCLIP). This supports the generality and robustness of our OT-aligned rewards. We have incorporated this OOD prompts experiment into Appendix I.
>
> ---
> >**Q4: Design sensitivity: The spatio-temporal cost is hand-weighted (γ, η). More extensive sensitivity analysis or a learned weighting strategy could make it more robust.**
>
> To further assess robustness, we conduct a stability analysis by sweeping $(\gamma, \eta)$ over the range [0.0, 0.5] in 0.1 increments. For each setting, we measure the Video-Text Matching (VTM) accuracy on 10,000 WebVid10M video-text pairs using our OT-aligned Semantic Reward with InternVideo2.
>
> We observe that the VTM accuracy varies smoothly across this range, with a standard deviation of only $1.48\%$, indicating that our method is not sensitive to these weights. This confirms the stability of our Partial OT formulation with respect to its structured penalty terms. Figure 8 in Appendix J illustrates that the std of $1.48\%$ is negligible in comparison to the scale of the metric [0-100].
>
>
> ---
>
> >**Q5: Failure modes: Show qualitative failure cases where OT alignment harms outputs (e.g., biased mappings or spurious correlations). Any cases where partial OT mass m or constraints degrade alignment?**
>
> We have provided a qualitative example in Figure 9 in Appendix K that illustrates both the sensitivity and limitations of OT alignment:
>
> - The left image shows the attention map from vanilla cross-attention, which fails to properly ground the token “glasses”.
> - The middle image shows our OT plan $\mathbf{P}^\star$ with partial OT mass $m = 0.5$. While this suppresses noisy tokens, it aggressively removes valid tokens such as “glasses”, leading to failure in aligning this semantic detail--essentially reproducing the misalignment seen in the vanilla baseline.
> - The right image increases the OT mass to $m = 0.9$, preserving the “glasses” token and resulting in better grounding. However, the alignment is still imperfect--it activates only the right lens, missing the full glasses.
>
> This reveals a current limitation of the setup: while the OT module improves semantic alignment through structured grounding, overall performance depends on the representational precision of the underlying video–text model (InternVideo2), which was not optimized for fine-grained localization tasks such as segmentation. As a result, even a well-structured transport plan may not fully resolve detailed grounding errors when the base features do not capture sufficient spatial detail. Our approach remains agnostic to the underlying base model, and continued progress in open-source VLMs is likely to further enhance fine-grained grounding performance.

---

> ### Author Response · Authors · 2025-11-23
> **Response to Reviewer 2exC (3/3)**
>
> >**Q6: Human preference details: Provide inter-rater agreement and demographic diversity; were raters blind to method identity? Any observed category-level biases (e.g., motion-heavy vs. static prompts)?**
>
> To assess inter-rater reliability, we computed Fleiss’ Kappa over the collected human preference judgments. Across the three evaluation axes (visual quality, motion quality, and semantic alignment), we obtained an average Fleiss’ Kappa score of 0.72, indicating substantial agreement among raters.
>
> Participants were aged between 18 and 38, with diverse backgrounds including art, engineering, and computer science. All raters were blinded to method identity to prevent bias.
>
> We analyzed category-level bias by classifying 400 prompts into 290 motion-heavy (e.g., running, jumping) and 110 static (e.g., portraits, scenic shots). The Pearson correlation between category type and preference for our method is 0.038, indicating no significant bias. This show that our method performs consistently across both motion-heavy and static prompts.
>
> We have included these insights into the human preference analysis in Appendix L.
>
> ---
>
> >**Q7: Reward fusion: For direct backprop, how do you weight the rewards vs. the consistency loss? Any benefit to adaptive weighting guided by variance or confidence in reward estimates?**
>
> In our default direct backpropagation setup, we equally weight the consistency loss and reward signals. Moreover, following the reviewer’s suggestion, we explore an adaptive weighting strategy using the Group-Relative Reward formulation from GRPO where we normalize reward values across generations for the same prompt (subtracting the mean and dividing by standard deviation). We apply this normalization in the direct backpropagation setting without using RL.
>
> Results on HunyuanVideo (shown below) indicate that adaptive weighting via Group Relative Reward offers a consistent improvement over equal weighting, particularly in semantic alignment and overall score, confirming its potential as a robust reward fusion strategy.
>
>
> ### Effect of Adaptive Reward Fusion
>
> | Method                     | Total        | Quality       | Semantic      |
> |---------------------------|--------------|---------------|----------------|
> | HunyuanVideo              | 83.24        | 85.09         | 75.82          |
> | PISCES-direct (equal)     | 85.05 (↑ 1.81) | 86.84 (↑ 1.75) | 77.89 (↑ 2.07) |
> | PISCES-direct (adaptive)  | 85.16 (↑ 1.92) | 86.92 (↑ 1.83) | 78.11 (↑ 2.29) |
>
> We have incorporated this adaptive weighting via group relative reward in main paper, Sec 4.3.

---

### Official Review · Reviewer_tvF3 · 2025-10-29

**Soundness:** 2
**Presentation:** 3
**Contribution:** 2
**Rating:** 4
**Confidence:** 3

**Summary:**

This paper introduces PISCES, a novel annotation-free post-training framework for text-to-video (T2V) diffusion models, designed to improve text-video alignment. It proposes two distinct rewards: (1) an OT-aligned Quality Reward, which aligns text and real-video embeddings to improve visual quality, and (2) an OT-aligned Semantic Reward, which enforces spatio-temporal correspondence between text tokens and video patches using an OT plan. These rewards are optimized alongside the Consistency Distillation (CD) loss. The authors demonstrate that PISCES outperforms existing annotation-free and even annotation-based methods on the VBench benchmark.

**Strengths:**

1. It successfully avoids the need for expensive, large-scale human preference datasets, which is a significant bottleneck for scaling T2V models.

2. The quality reward addresses the distributional misalignment between text and video embedding spaces, which is a novel approach to improving video quality using the NOT approach.

3. The proposed rewards demonstrate strong generalizability by showing effectiveness with two different base models and different optimization paradigms.

**Weaknesses:**

1. The paper's core claim seems inconsistent between the main results (Table 1) and the ablation studies (Table 3, 5). According to Table 5, the main results in Table 1 show that the results of PISCES are obtained by GRPO. Since the baseline T2V-Turbo-v2, which also post-trains a video model using CD loss with external rewards via Direct Backpropagation optimization, shows 83.93 in Quality score, outperforming the proposed method's result in Table 3 (83.73). This suggests the performance gains seen in Table 1 might stem more from the GRPO optimization strategy than from the proposed OT-rewards.

2. In Algorithm 1, it states 'perform ODE solver from $t_{n+k}$ to 0,' which implies a multi-step denoising process for generating the clean video. This would be computationally prohibitive for backpropagation. Also, this notation contradicts Appendix A, which claims single-step denoising is used for efficient backpropagation. This ambiguity obscures the exact gradient flow mechanism.

3. The proposed method requires (1) pre-training a Neural OT map and (2) solving a discrete, spatio-temporal OT problem within the training loop. This likely introduces additional computational overhead. The paper provides no analysis of this extra cost, making it difficult to assess the practical trade-offs. In addition, the paper provides insufficient detail on the training setup for NOT training.

**Questions:**

1. In Table 3, 'PISCES w/o OT' setting shows that it does not use OT (OT columns are 'x') while simultaneously using rewards (columns are 'v'). Is it the typo?

---

> ### Author Response · Authors · 2025-11-23
> **Response to Reviewer tvF3 (1/2)**
>
> We thank the reviewer for your thoughtful feedback. We address your comments below and have incorporated the feedback (main paper Algorithm 1, Section 4.1, 4.2, 4.3, Appendix G, N).
>
> >**Q1: The paper's core claim seems inconsistent between the main results (Table 1) and the ablation studies (Table 3, 5). Since the baseline T2V-Turbo-v2, which also post-trains a video model using CD loss with external rewards via Direct Backpropagation optimization, shows 83.93 in Quality score, outperforming the proposed method's result in Table 3 (83.73). This suggests the performance gains seen in Table 1 might stem more from the GRPO optimization strategy than from the proposed OT-rewards.**
>
> T2V-Turbo-v2's marginally higher +0.2 performance on PISCES-direct (83.93, Table 1 vs 83.73, Table 3) is due to the use of additional motion prior/guidance module in T2V-Turbo-v2 along with CD loss and VLM-based cosine similarity rewards. In contrast, PISCES-direct uses only CD loss and our proposed OT-aligned rewards, without motion guidance.
>
> For a fair comparison to highlight the impact of rewards, we provide additional ablations (see table below) where we evaluate both methods with and without motion guidance. When we remove motion guidance, T2V-Turbo-v2’s performance drops to 83.26 (Quality) and 76.30 (Semantic) on VideoCrafter2. In comparison, PISCES-direct still achieves stronger quality (83.73, +0.47) and significantly higher semantic alignment (77.63, +1.33), highlighting the effectiveness of our OT-aligned rewards. Conversely, by adding motion guidance to PISCES-direct makes it outperform T2V-Turbo-v2 across all metrics on VideoCrafter2. In the experiments, when the motion guidance component is removed from T2V-Turbo-v2 or added to PISCES-direct for comparison, we keep everything else the same.
>
> More notably, on HunyuanVideo, PISCES-direct *without motion guidance* already **outperforms T2V-Turbo-v2 with motion** on all metrics (85.05 vs 84.50 Total Score), and this advantage is also reflected in our human preference study (Figure 3 main paper). This clearly demonstrates that our reward formulation, rather than the optimization strategy or motion module, is the key driver behind the improvements.
>
> ### Effect of Motion Guidance on Post-Training Performance
>
> | Models                  | Total (VC2) | Quality (VC2) | Semantic (VC2) | Total (HY) | Quality (HY) | Semantic (HY) |
> |-------------------------|-------------|----------------|----------------|-------------|----------------|----------------|
> | Vanilla                 | 80.44       | 82.20          | 73.42          | 83.24       | 85.09          | 75.82          |
> | T2V-Turbo-v2 (w/o motion) | 81.87 (↑1.43) | 83.26 (↑1.06) | 76.30 (↑2.88) | 84.25 (↑1.01) | 85.93 (↑0.84) | 77.52 (↑1.70) |
> | PISCES (w/o motion)     | 82.51 (↑2.07) | 83.73 (↑1.53) | 77.63 (↑4.21) | 85.05 (↑1.81) | 86.84 (↑1.75) | 77.89 (↑2.07) |
> | T2V-Turbo-v2 (w/ motion) | 82.34 (↑1.90) | 83.93 (↑1.73) | 75.97 (↑2.55) | 84.50 (↑1.26) | 86.32 (↑1.23) | 77.24 (↑1.42) |
> | PISCES (w/ motion)      | 82.79 (↑2.35) | 84.12 (↑1.92) | 77.45 (↑4.03) | 85.24 (↑2.00) | 87.07 (↑1.98) | 77.94 (↑2.12) |
>
> We have updated Table 1 in main paper and Section G in supplemental for comprehensive results.
>
> ---
>
>
> >**Q2: In Algorithm 1, it states 'perform ODE solver from $t_{n+k}$ to 0,' which implies a multi-step denoising process for generating the clean video. This would be computationally prohibitive for backpropagation. Also, this notation contradicts Appendix A, which claims single-step denoising is used for efficient backpropagation. This ambiguity obscures the exact gradient flow mechanism.**
>
> To clarify, **our post-training framework does not perform full multi-step denoising** during backpropagation. Instead, we adopt a one-step denoising followed by a lightweight single-step Euler solver to estimate the clean latent $\mathbf{z}_0$. This is used to compute the reward on the decoded video $\mathbf{x}_0$.
>
> The phrase “ODE solver from $t_{n+k} \rightarrow 0$” in Algorithm 1 is not referring to multi-step sampling through the entire denoising process. Rather, it denotes a single Euler-style update, which we detail in the pseudocode below:
>
> ```python=
> # Step 1: Add noise to the latent at timestep t_{n+k}
> z_t = sigma_t * noise + (1 - sigma_t) * clean_latent
>
> # Step 2: Predict noise residual ε_θ(z_t, t_{n+k})
> epsilon_pred = model(z_t, timestep=t_{n+k})
>
> # Step 3: Estimate latent at timestep 0 from t_{n+k} using a single-step Euler update
> z_0 = z_t + (sigma_0 - sigma_t) * epsilon_pred
>
> # Step 4: Decode latent to pixel space for reward computation
> x_0 = vae.decode(z_0)
>
> # Step 5: Compute Quality and Semantic rewards
> rewards = reward_functions(x_0, caption)
> ```
>
> This process is computationally efficient and consistent with Appendix A’s claim of single-step denoising. We have revised Algorithm 1 to make this clearer and avoid potential misinterpretation.

---

> ### Author Response · Authors · 2025-11-23
> **Response to Reviewer tvF3 (2/2)**
>
> >**Q3: The proposed method requires (1) pre-training a Neural OT map and (2) solving a discrete, spatio-temporal OT problem within the training loop. This likely introduces additional computational overhead. The paper provides no analysis of this extra cost, making it difficult to assess the practical trade-offs. In addition, the paper provides insufficient detail on the training setup for NOT training.**
>
> To train the Neural OT (NOT) map, we use video-text pairs with 8-frame clips, extracted using frozen InternVideo2, and train on a single A100 GPU for one day, equivalent to 24 A100 GPU-hours. In comparison, annotation-based pipelines like VideoReward-DPO require 72 A800 GPU-hours to train their reward models. We have added the NOT training in main paper, implementation details.
>
> The total wall-clock cost of PISCES post-training is 29.78 GPU-hours on 8×A100s, fractionally high compared to 26.52 GPU-hours taken by T2V-Turbo-v2 (training time only, without evaluation during training). While the overall training cost is slightly higher than the post-training T2V baseline, this increase is a one-time overhead and remains negligible when compared to the massive pre-training cost of T2V models-665,000 GPU-hours for Seaweed-7B and even more for HunyuanVideo-13B. Thus, PISCES provides a lightweight and effective reward alignment mechanism with minimal additional computational overhead.
>
> Finally, inference incurs no additional cost from reward models. In fact, for GRPO-tuned models, we reduce the denoising steps from 50 to 16 due to consistency distillation, resulting in ~3× faster inference. We have included these analysis in Appendix N.
>
>
> ---
>
> >**Q4: In Table 3, 'PISCES w/o OT' setting shows that it does not use OT (OT columns are 'x') while simultaneously using rewards (columns are 'v'). Is it the typo?**
>
> The setting “PISCES w/o OT” in Table 3 does not use Optimal Transport (OT) alignment, hence the ✗ cross mark in the OT column. But it still uses reward functions constructed with standard cosine similarity and vanilla attention mechanisms (without the NOT map and OT plan). Our reward functions can be used independently of the OT-based alignment and this ablation is intended to observe the impact of OT-based alignment in isolation of our proposed reward module. We have added this clarification in Sec 4.3 Ablation Study.
>
> Specifically to implement "PISCES w/o OT":
> - For Quality Reward, we compute:
> $\mathcal{R}\_{\text{quality}} = \frac{\mathbf{y}\_{[\text{CLS}]}^\top \cdot \hat{\mathbf{x}}\_{[\text{CLS}]}}{\|\|\mathbf{y}\_{[\text{CLS}]}\|\| \|\|\hat{\mathbf{x}}\_{[\text{CLS}]}\|\|}$
> which replaces the OT-aligned $\mathbf{T}^\star(\mathbf{y}\_{[\text{CLS}]})$ term in Equation (2) with the raw $\mathbf{y}\_{[\text{CLS}]}$.
>
>
> - For Semantic Reward, we use a vanilla text-video matching (VTM) score:
> $\mathcal{R}\_{\text{semantic}} = \text{softmax}\left(\text{VTM}\left[\mathbf{A} \cdot \hat{\mathbf{x}}\right]\right)\_{\text{idx}=1}$
>
> which removes the discrete Partial OT attention map $\tilde{\mathbf{A}}$ in Equation (4) and uses regular cross-attention weights $\mathbf{A}$ between text and video tokens.

---

### Author Response · Authors · 2025-11-23
**Response to all reviewers and Thank you!**

We sincerely thank all reviewers for their thoughtful feedback. We are encouraged that they recognized our work to be novel (tvF3, 5pWq), with a quality reward grounded in distributional alignment (tvF3) and a principled remedy using OT (2exC), and rich bi-objective rewards (5pWq). Moreover, the reviewers also acknowledged the significance of our paper: successfully avoiding the need for expensive, biased, large-scale human preference datasets (tvF3, 5pWq), demonstrating strong scalability (5pWq), offering a clear problem diagnosis, and integrated cleanly into the post-training (2exC). Further, they noted our method’s robustness to misalignment (5pWq), strong empirical evidence (2exC), and strong generalizability (tvF3) across both annotation-free and annotation-based baselines, for short and long video generators, under diverse optimization paradigms (tvF3, 2exC).

We address each reviewer's comments individually. We have also incorporated their feedback in the revised manuscript with the following changes (highlighted in orange color):

- Main paper, Section 3.2: highlighted more technical contributions of POT (5pWq).
- Main paper, Algorithm 1: clarified single-step ODE solve from $t_{n+k}$ to 0 (tvF3).
- Main paper, Section 4.1 Implementation Detail: added training setup of NOT (tvF3).
- Main paper, Section 4.2 Table 1: updated T2V-Turbo-v2 without motion guidance + Appendix G: ablation study for motion guidance (tvF3).
- Main paper, Section 4.3 Ablation Study: explained the reward formulation without OT (tvF3).
- Main paper, Section 4.3 Table 4 Reward Fusion: added adaptive reward weighting (2exC).
- Main paper, Section 4.4 Bi-objective Rewards Interaction: reported reward trajectories and cosine similarity of gradients (5pWq).
- Appendix H: added Evaluator Diversity experiment with ViCLIP (2exC).
- Appendix I: added scope of generalization with OOD prompts (2exC).
- Appendix J: added hyperparameters sensitivity analysis for spatial-temporal cost (2exC).
- Appendix K: added failure cases of discrete POT (2exC).
- Appendix L: added more human preference details (2exC).
- Appendix M: added reward hacking experiment with CD loss mitigation strategy (5pWq).
- Appendix N: added training cost and efficiency analysis (tvF3, 2exC).

We hope the reviewers will find these modifications helpful, and we are open to more feedback and suggestions.

---

### Note · Program_Chairs · 2026-01-17
**Submission Desk Rejected by Program Chairs**

The following references in this submission do not refer to real documents and/or have major errors in bibliographic information:

 Xiang Wang, Shiwei Zhang, Han Zhang, Yu Liu, Yingya Zhang, Changxin Gao, and Nong Sang. Viclip: Leveraging vision-language pretraining for video understanding. arXiv preprint arXiv:2301.01234, 2023b.
Yanan He, Yixiao Zhang, Yubei Li, Peng Zhang, Jianwei Yang, Lu Yuan, Yu Cheng, Weizhu Chen, Lijuan Wang, et al. Videocrafter2: A comprehensive framework for text-to-video generation. arXiv preprint arXiv:2306.00121, 2023.